# WORLDMIRROR: UNIVERSAL 3D WORLD RECONSTRUCTION WITH ANY-PRIOR PROMPTING

## ABSTRACT

We present *WorldMirror*, an all-in-one, feed-forward model for versatile 3D geometric prediction tasks. Unlike existing methods constrained to image-only inputs or customized for a specific task, our framework flexibly integrates diverse geometric priors, including camera poses, intrinsics, and depth maps, while simultaneously generating multiple 3D representations: dense point clouds, multi-view depth maps, camera parameters, surface normals, and 3D Gaussians. This elegant and unified architecture leverages available prior information to resolve structural ambiguities and delivers geometrically consistent 3D outputs in a single forward pass. *WorldMirror* achieves state-of-the-art performance across diverse benchmarks from camera, point map, depth, and surface normal estimation to novel view synthesis, while maintaining the efficiency of feed-forward inference. Code and models will be publicly available.

Figure 1: *WorldMirror* is a large feed-forward 3D reconstruction model that takes raw images along with optional priors (depth, calibrated intrinsics, camera pose) as input and produces high-quality geometric attributes in seconds, including point clouds, 3DGS, cameras, depth, and normal maps.

## 1 INTRODUCTION

Visual geometry learning is a fundamental problem in computer vision, with applications spanning augmented reality, robotics, and autonomous navigation. Traditional Structure-from-Motion (SfM) (Schonberger & Frahm, 2016) and Multi-View Stereo (MVS) algorithms rely on iterative optimization, making them computationally expensive. The field has recently shifted toward feed-forward neural networks that directly reconstruct geometry from visual inputs. These end-to-end models, exemplified by DUSt3R (Wang et al., 2024) and its successors, have demonstrated remarkable capabilities in processing image pairs, videos, and multi-view images.

Despite significant progress, existing methods still face two critical limitations regarding their input and output spaces. On the input front, these approaches exclusively process raw images, failing to leverage additional modalities that are useful and often accessible in real-world applications, such as calibrated camera intrinsics, camera poses, and depth measurements derived from LIDAR or RGB-D

sensors. Without incorporating these prior cues, current methods encounter unnecessary challenges in scenarios that could otherwise be readily addressed: calibrated intrinsics resolve scale ambiguities, camera poses ensure multi-view consistency, and depth measurements ground predictions in areas where image-based cues alone are insufficient, such as textureless or reflective regions.

Second, existing methods are typically limited to addressing single or limited tasks in output space. These approaches are often highly specialized, *e.g.*, focusing on depth estimation (Yang et al., 2024), point map regression (Wang et al., 2024), camera pose prediction (Wang et al., 2023a), or point tracking (Karaev et al., 2024), and rarely integrate multiple tasks within a unified framework. Recently, VGGT (Wang et al., 2025a) has explored unifying these tasks, but some fundamental geometry tasks like surface normal estimation and novel view synthesis remain excluded. These two limitations prompt a critical question: can we reconcile both challenges by effectively leveraging diverse prior knowledge within a universal 3D reconstruction architecture?

To address these challenges, we introduce *WorldMirror*, a framework designed to perform universal 3D reconstruction tasks while leveraging any available geometric priors. At the core of *WorldMirror* is a novel **Multi-Modal Prior Prompting** mechanism that embeds diverse prior modalities, including calibrated intrinsics, camera pose, and depth, into the feed-forward model. Given any subset of the available priors, we utilize several lightweight encoding layers to convert each modality into structured tokens. Rather than treating all prior modalities uniformly, we implement specialized embedding strategies for each modality type. Camera poses and calibrated intrinsics are encoded into a single token due to their compact nature. Depth maps, rich in spatial information, are converted to dense tokens. These tokens maintain spatial alignment with visual tokens and are integrated through direct addition. Furthermore, to reduce the training-inference gap, we propose a dynamic prior injection scheme by randomly sampling distinct prior combinations during training, enabling the model to adapt to arbitrary subsets (including none) of available priors during inference.

Besides, *WorldMirror* features a **Universal Geometric Prediction** architecture capable of handling the full spectrum of 3D reconstruction tasks from camera and depth estimation to point map regression, surface normal estimation, and novel view synthesis. *WorldMirror* builds upon a fully transformer-based architecture for regressing camera parameters and uses unified decoder heads for all other dense prediction tasks. Incorporating these tasks together broadens the model's capabilities toward a versatile 3D reconstruction framework. However, training such a multi-task 3D reconstruction foundation model poses significant challenges, as geometric quantities are inherently coupled and require carefully designed training strategies. We thus propose a systematic curriculum learning strategy to optimize training efficiency and enhance performance by progressing from simple to complex across three dimensions: task sequencing, data scheduling, and progressive resolution.

Extensive experiments demonstrate that *WorldMirror* achieves state-of-the-art performance across diverse benchmarks and tasks. It surpasses recent 3D reconstruction methods, such as VGGT (Wang et al., 2025a) and $\pi^3$ (Wang et al., 2025c) in point map and camera estimation, while outperforming StableNormal (Ye et al., 2024b) and GeoWizard (Fu et al., 2024) in surface normal prediction and significantly exceeding recent method AnySplat (Jiang et al., 2025) in novel view synthesis.

We summarize our contributions as follows: 1) We propose a universal 3D world reconstruction model capable of taking multi-modal priors as guidance, including per-view calibrated intrinsics, camera pose, and depth maps. 2) Our model serves as a foundational 3D reconstruction framework, which supports universal geometric predictions from point map, camera, depth, and surface normal estimation to novel view synthesis. 3) Extensive experiments show that our method outperforms existing methods across diverse tasks qualitatively and quantitatively.

## 2 RELATED WORKS

**Feed-Forward 3D Reconstruction.** Feed-forward 3D reconstruction models have recently emerged as powerful alternatives to traditional SfM/MVS pipelines by directly regressing 3D structure. DUSt3R (Wang et al., 2024) pioneers this direction with point map prediction, while Fast3R (Yang et al., 2025) improves its scalability. VGGT (Wang et al., 2025a) further introduces large-scale multi-task learning, with subsequent variants that remove reference-view bias (Wang et al., 2025c) and extend to kilometer-scale sequences (Deng et al., 2025). Meanwhile, Dens3R (Fang et al., 2025) introduces a dense prediction backbone for joint estimation of geometric attributes. Building

on these advances, *WorldMirror* unifies an even broader range of 3D tasks, including camera poses, depth, surface normals, point maps, and novel view synthesis, in one feed-forward pass.

**3D Prior Guidance.** Traditional optimization-based methods like COLMAP (Schönberger et al., 2016) incorporate known camera parameters to improve reconstruction quality. Recent learning-based approaches have also explored different forms of guidance: UniDepth (Piccinelli et al., 2024) optionally uses camera intrinsics for improved monocular depth estimation, while some video diffusion models (He et al., 2024; Huang et al., 2025; Team, 2025) demonstrate how camera trajectories can guide consistent content generation. More recently, Pow3R (Jang et al., 2025) extends DUSt3R (Wang et al., 2024) with additional modalities as input but remains limited to sparse-view inputs within the "3R" paradigms. The integration of more modalities into dense regression frameworks like VGGT remains unexplored. In this paper, we present the first systematic exploration of multi-modal geometric prior injection within dense multi-view reconstruction frameworks.

**Multi-task Learning.** Multi-task learning (MTL) has been extensively studied for dense prediction tasks in computer vision. Early works (Kendall et al., 2018) proposed uncertainty-based weighting to balance multiple task losses, while Taskonomy (Zamir et al., 2018) systematically explored task relationships and transfer learning structures. Pattern-structure diffusion (Zhou et al., 2020) further investigated how to effectively share representations across related tasks. These methods have demonstrated that joint training on multiple tasks can improve generalization and computational efficiency through shared feature learning. Unlike prior 2D multi-task works where standard joint training succeeds, we discover that 3D Gaussian Splatting heads (optimizing for photo-realistic rendering) conflict with dense geometry prediction heads (optimizing for precise geometry), and address this through a curriculum learning strategy that trains geometry tasks first, then trains only the GS head with frozen geometry features.

**Generalizable Novel View Synthesis.** Novel view synthesis (NVS) has been extensively studied with representations such as NeRF (Mildenhall et al., 2021) and 3D Gaussian Splatting (Kerbl et al., 2023), which achieve photorealistic results but typically require dense-view training for each scene. Early generalizable NVS methods (Yu et al., 2021; Charatan et al., 2024; Xu et al., 2025b; Liu et al., 2025) take sparse-view images with known intrinsics and poses as input to produce 3D scenes or novel views. While effective for sparse inputs, these approaches depend on accurate calibration or fixed view counts (Chen et al., 2024; Min et al., 2024). Pose-free methods (Jiang et al., 2023; Wang et al., 2023b; Ye et al., 2024a) instead pursue end-to-end reconstruction directly from images. FLARE (Zhang et al., 2025) introduces a cascaded pose-geometry-appearance pipeline, while AnySplat (Jiang et al., 2025) combines 3D foundation models with 3D Gaussians for real-time NVS from uncalibrated images. We advance beyond these methods by enabling pose-free novel view synthesis with flexible input view counts, optional prior incorporation, and superior rendering quality.

## 3 METHOD

Given $N$ multi-view images $\{\boldsymbol{I}_i\}_{i=1}^{N}$, our work aims to utilize any available priors for unified geometric predictions. To this end, we introduce *multi-modal prior prompting* (Sec. 3.1) to embed priors including calibrated intrinsics, camera poses, and depth maps seamlessly into dense visual tokens as guidance for our model. To unify various geometric predictions, we present *universal geometric prediction* (Sec. 3.2) to predict various geometric attributes, including point maps, multi-view depths, camera parameters, surface normals, and 3D Gaussians, within our unified framework. To reduce the training-inference gap and achieve the optimal overall performance, we introduce a dynamic prior injection scheme with well-designed curriculum learning strategies (Sec. A.2).

### 3.1 MULTI-MODAL PRIOR PROMPTING

As demonstrated in previous works (Piccinelli et al., 2024; Jang et al., 2025), auxiliary information like calibrated intrinsics, depths, and camera poses substantially enhances visual geometric learning. This motivates us to develop a model that flexibly leverages available priors when present, while maintaining robust reconstruction quality when priors are unavailable. In the following, we discuss how to effectively embed diverse modality information as input to our model, and then describe the training strategy that enables the model to flexibly infer with any priors.

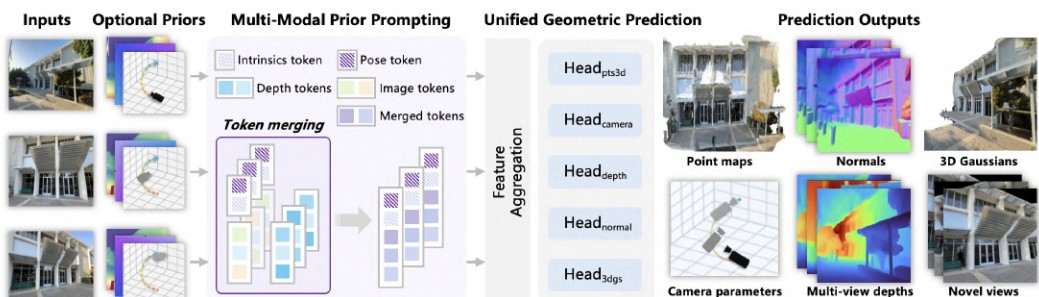

Figure 2: **Overview of *WorldMirror*.** Given multi-view images with optional priors (depths, calibrated intrinsics, camera poses) as input, our framework encodes each prior modality into tokens and integrates them with image tokens. The composite tokens are subsequently processed by a visual transformer backbone to effectively aggregate multi-view features. The consolidated representations are then passed to multi-task heads to generate comprehensive geometric outputs, including point maps, camera parameters, multi-view depth maps, surface normals, and 3D Gaussians.

**Camera Pose.** Given the camera poses $\{[\boldsymbol{R}_i|\boldsymbol{t}_i]\}_{i=1}^N$ of input images, where $\boldsymbol{R}_i \in \mathbb{R}^{3\times3}, \boldsymbol{t}_i \in \mathbb{R}^3$, we first normalize the scene scale to a standard unit cube, and the new translation vector $\boldsymbol{t}^{norm}$ is formulated as: $\boldsymbol{t}_i^{norm} = (\boldsymbol{t}_i - \boldsymbol{c})/\alpha$, where $\boldsymbol{c}$ is the camera center and $\alpha$ is the maximum distance of each camera to $\boldsymbol{c}$. This normalization ensures consistent numerical ranges regardless of the scene scale. Then, to integrate camera information, we encode each camera pose $[\boldsymbol{R}_i|\boldsymbol{t}_i^{norm}]$ into a single token due to their compact representation. Specifically, we convert each rotation matrix $\boldsymbol{R}_i \in \mathbb{R}^{3\times3}$ to a quaternion $\boldsymbol{q}_i \in \mathbb{R}^4$ and combine it with the normalized translation vector $\boldsymbol{t}_i^{norm} \in \mathbb{R}^3$ to form a 7-dimensional vector. This vector is then projected to $\boldsymbol{T}_i^{cam} \in \mathbb{R}^{1\times D}$ using a two-layer MLP, where $D$ matches the dimension of image tokens, enabling seamless token concatenation.

**Calibrated Intrinsics.** Embedding calibrated camera intrinsics is comparatively straightforward. Given the intrinsic matrix $\boldsymbol{K}_i \in \mathbb{R}^{3\times3}$ of each image, we extract the focal lengths and principal points $(f_x, f_y, c_x, c_y)$ and normalize them by dividing the image width $W$ and height $H$, respectively. This normalization ensures training stability across images with varying resolutions. Similar to camera pose, we project the normalized intrinsic to $\boldsymbol{T}_i^{intr} \in \mathbb{R}^{1\times D}$ using a two-layer MLP, enabling seamless concatenation with visual tokens.

**Depth Map.** Unlike camera poses and intrinsics that are compact representations, depth maps are dense spatial signals requiring different embedding strategies. Given a depth map $\boldsymbol{D}_i \in \mathbb{R}^{H\times W}$, we first normalize its values to the range $[0, 1]$ to ensure numerical stability. Then, we employ a convolutional layer with kernel size matching the patch size used for visual tokens to create depth tokens $\boldsymbol{T}_i^{depth} \in \mathbb{R}^{(H_p \times W_p)\times D}$, where $H_p, W_p$ are the token height and width, respectively. These depth tokens are spatially aligned with the visual tokens and are directly added to them. This additive integration preserves the spatial structure of the scene while enriching visual tokens with geometric information, fusing appearance and geometry in a unified representation.

**Versatile Prior Prompting.** To enable versatile prior-prompted 3D reconstruction, we concatenate intrinsics tokens and camera pose tokens with image tokens $\boldsymbol{T}_i^{img} \in \mathbb{R}^{(H_p \times W_p)\times D}$, while directly adding depth tokens, resulting in a prompted token set $\boldsymbol{T}_i^{prompt}$ as:

$$\boldsymbol{T}_i^{prompt} = [\boldsymbol{T}_i^{cam}, \ \boldsymbol{T}_i^{intr}, \boldsymbol{T}_i^{img} + \boldsymbol{T}_i^{depth}], \quad \boldsymbol{T}_i^{prompt} \in \mathbb{R}^{(1+1+H_p \times W_p)\times D} \tag{1}$$

Considering that during inference, we may not have access to all modality information, we thus propose a dynamic prior injection scheme during training, which allows the model to adapt to arbitrary combinations of priors, as stated in Sec. A.2.

## 3.2 UNIVERSAL GEOMETRIC PREDICTION

Recent approaches, such as VGGT, have unified various geometry prediction tasks, but lack support for some common applications like novel view synthesis and surface normal estimation. In

this work, we propose a more comprehensive framework enabling universal geometric prediction, including point maps, camera parameters, depth maps, surface normals, and 3D Gaussians.

**Point Map, Camera, and Depth Estimation.** Following the design of VGGT, given the output tokens $\boldsymbol{T}_i^{out} \in \mathbb{R}^{L \times D}$ of visual transformer backbone, we utilize DPT heads $\text{DPT}(\cdot)$ (Ranftl et al., 2021) to regress dense outputs, including 3D point map $\hat{\boldsymbol{P}}_i$ and multiview depth $\hat{\boldsymbol{D}}_i$, and use transformer layers to predict camera parameters $\hat{\boldsymbol{E}}_i$ from camera tokens:

$$\hat{\boldsymbol{P}}_i = \text{DPT}_p(\hat{\boldsymbol{T}}_i^{img}), \quad \hat{\boldsymbol{D}}_i = \text{DPT}_d(\hat{\boldsymbol{T}}_i^{img}), \quad \hat{\boldsymbol{E}}_i = \text{Transformer}(\hat{\boldsymbol{T}}_i^{cam}) \tag{2}$$

**Surface Normal Estimation.** For surface normal estimation, we employ the same DPT architecture as other dense prediction tasks, followed by L2 normalization to ensure unit vector outputs:

$$\hat{\boldsymbol{N}}_i = \text{DPT}_n(\hat{\boldsymbol{T}}_i^{img}) \,/\, ||\text{DPT}_n(\hat{\boldsymbol{T}}_i^{img})||_2. \tag{3}$$

To address the scarcity of ground-truth normal annotations, we introduce a hybrid supervision approach. We leverage both annotated datasets and pseudo normals derived from ground-truth depth maps via plane fitting for datasets lacking normal labels, which enables effective usage of diverse data for generalization while ensuring consistent normal estimation.

**Novel View Synthesis.** To enable novel view synthesis, we predict 3D Gaussian Splatting (3DGS). Specifically, we use a DPT head $\text{DPT}_g(\cdot)$ to regress pixel-wise Gaussian depth maps $\hat{\boldsymbol{D}}_g$ and Gaussian feature maps $F_g$. These depth predictions are back-projected using the ground-truth camera poses $[\boldsymbol{R}|\boldsymbol{t}]$ and intrinsic matrix $\boldsymbol{K}$ to obtain the Gaussian centers $\boldsymbol{\mu}_g$. To infer the remaining Gaussian attributes $\hat{\boldsymbol{G}}$, including opacity $\sigma_g$, orientation $r_g$, scale $s_g$, residual spherical-harmonic color coefficients $\Delta \mathbf{c}_g$, and a fusion weight $w_g$, we combine $F_g$ with appearance features derived from a convolution network $\text{Conv}(\cdot)$. The overall process can be formulated as:

$$\hat{\boldsymbol{G}} = \text{Conv}(F_g, \boldsymbol{I}), \qquad \hat{\boldsymbol{D}}_g, F_g = \text{DPT}_g(\hat{\boldsymbol{T}}^{img}) \tag{4}$$

To reduce Gaussian redundancy caused by overlapping regions across multiple views, we cluster and prune per-pixel Gaussians through voxelization, similar to AnySplat (Jiang et al., 2025). To enable novel view synthesis, the input images are split into context and target sets during training. The 3D Gaussians are built only from context views but rendered to and supervised by both target and original context viewpoints via a differentiable rasterizer (Ye et al., 2025). This dual supervision enables the model to synthesize novel views while preserving consistency with input observations.

**Training Losses.** Our model is trained end-to-end by minimizing a composite loss function, $\mathcal{L}$, which integrates supervision for all prediction tasks as:

$$\mathcal{L} = \mathcal{L}_{\text{points}} + \mathcal{L}_{\text{depth}} + \mathcal{L}_{\text{cam}} + \mathcal{L}_{\text{normal}} + \mathcal{L}_{\text{3dgs}} \tag{5}$$

Please refer to Sec. A.1 for the details of training losses.

## 4 EXPERIMENTS

In this section, we evaluate our approach across four tasks (Sec. 4.1): point map reconstruction, camera pose estimation, surface normal estimation, and novel view synthesis. We also evaluate the effectiveness of different configurations of input priors with a prior-guidance benchmark (Sec. 4.2), and conduct an ablation study to evaluate our design choices (Sec. 4.4). To demonstrate the generalization ability of our method with in-the-wild inputs, we predict the 3D Gaussians (Fig. 8) and point clouds (Fig. 10) with diverse styles of AI-created videos. Details of training settings and data usage can be found in Sec. A.2.

### 4.1 EVALUATION ON DIFFERENT TASKS

**Point Map Reconstruction.** We assess point map reconstruction quality across both scene-level and object-level datasets: 7-Scenes (Shotton et al., 2013), NRGBD (Azinović et al., 2022), and DTU (Jensen et al., 2014). We use multi-view images with fixed sequence-id mappings from (Wang et al., 2025c) for fair comparison, reporting Accuracy (Acc.) and Completion (Comp.) metrics in Tab. 1. Our method without any priors already surpasses previous SOTA approaches VGGT and

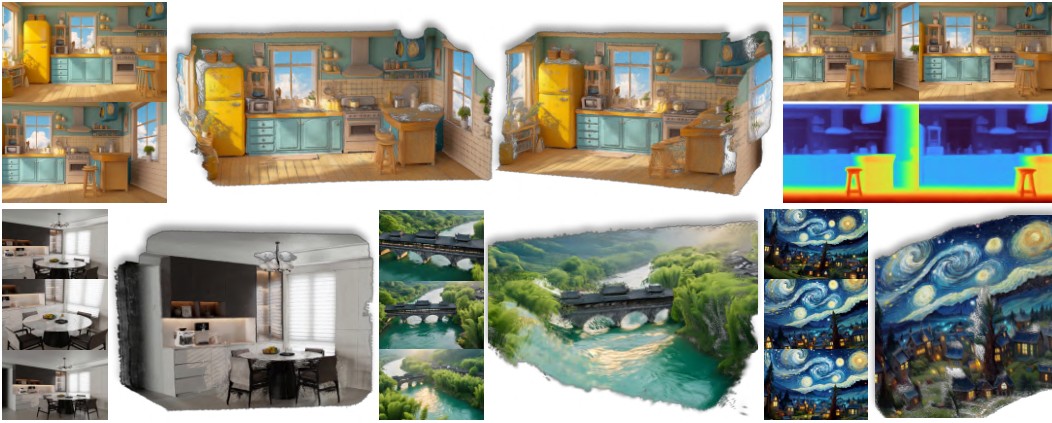

Figure 3: **Feed-Forward 3D Gaussians Predicted by *WorldMirror* with In-The-Wild Inputs.** Besides real photos, our method generalizes well to AI-created videos spanning diverse styles.

Table 1: **Point map Reconstruction on 7-Scenes, NRGBD, and DTU.** We report the performance of WorldMirror under different input configurations. The best results are **bold**.

| Method | 7-Scenes (scene) | | | | NRGBD (scene) | | | | DTU (object) | | | |
| | Acc. ↓ | | Comp. ↓ | | Acc. ↓ | | Comp. ↓ | | Acc. ↓ | | Comp. ↓ | |
| | Mean | Med. | Mean | Med. | Mean | Med. | Mean | Med. | Mean | Med. | Mean | Med. |
|---|---|---|---|---|---|---|---|---|---|---|---|---|
| Fast3R (Yang et al., 2025) | 0.096 | 0.065 | 0.145 | 0.093 | 0.135 | 0.091 | 0.163 | 0.104 | 3.340 | 1.919 | 2.929 | 1.125 |
| CUT3R (Wang et al., 2025b) | 0.094 | 0.051 | 0.101 | 0.050 | 0.104 | 0.041 | 0.079 | 0.031 | 4.742 | 2.600 | 3.400 | 1.316 |
| FLARE (Zhang et al., 2025) | 0.085 | 0.058 | 0.142 | 0.104 | 0.053 | 0.024 | 0.051 | 0.025 | 2.541 | 1.468 | 3.174 | 1.420 |
| VGGT (Wang et al., 2025a) | 0.046 | 0.026 | 0.057 | 0.034 | 0.051 | 0.029 | 0.066 | 0.038 | 1.338 | 0.779 | 1.896 | 0.992 |
| $\pi^3$(Wang et al., 2025c) | 0.048 | 0.028 | 0.072 | 0.047 | 0.026 | 0.015 | 0.028 | 0.014 | 1.198 | 0.646 | 1.849 | 0.607 |
| WorldMirror | 0.043 | 0.026 | 0.049 | 0.028 | 0.041 | 0.020 | 0.045 | 0.019 | 1.017 | 0.564 | 1.780 | 0.690 |
| WorldMirror (w/ intrinsics) | 0.042 | 0.028 | 0.048 | 0.026 | 0.041 | 0.020 | 0.045 | 0.019 | 0.977 | 0.542 | 1.762 | 0.682 |
| WorldMirror (w/ depth) | 0.038 | 0.024 | 0.039 | 0.023 | 0.032 | 0.015 | 0.031 | 0.014 | 0.831 | 0.506 | 1.022 | 0.599 |
| WorldMirror (w/ camera pose) | 0.023 | 0.014 | 0.036 | 0.019 | 0.029 | 0.018 | 0.032 | 0.017 | 0.990 | 0.548 | 1.847 | 0.686 |
| WorldMirror (w/ intrinsics/depth/camera pose) | **0.018** | **0.011** | **0.023** | **0.014** | **0.016** | **0.011** | **0.014** | **0.010** | **0.735** | **0.461** | **0.935** | **0.550** |

Table 2: **Camera Pose Estimation on RealEstate10K, Sintel, and TUM-dynamics.** All datasets are excluded from the training set, except that RealEstate10K was included for CUT3R training.

| Method | RealEstate10K (mixed, static) | | | Sintel (outdoor, dynamic) | | | TUM-dynamics (indoor, dynamic) | | |
| | RRA@30 ↑ | RTA@30 ↑ | AUC@30 ↑ | ATE↓ | RPE trans↓ | RPE rot↓ | ATE↓ | RPE trans↓ | RPE rot↓ |
|---|---|---|---|---|---|---|---|---|---|
| Fast3R(Yang et al., 2025) | 99.05 | 81.86 | 61.68 | 0.371 | 0.298 | 13.75 | 0.090 | 0.101 | 1.425 |
| CUT3R (Wang et al., 2025b) | 99.82 | 95.10 | 81.47 | 0.217 | 0.070 | 0.636 | 0.047 | 0.015 | 0.451 |
| FLARE (Zhang et al., 2025) | 99.69 | 95.23 | 80.01 | 0.207 | 0.090 | 3.015 | 0.026 | 0.013 | 0.475 |
| VGGT (Wang et al., 2025a) | 99.97 | 93.13 | 77.62 | 0.167 | 0.062 | 0.491 | 0.012 | 0.010 | 0.312 |
| $\pi^3$ (Wang et al., 2025c) | **99.99** | 95.62 | 85.90 | **0.074** | **0.040** | **0.282** | 0.014 | **0.009** | 0.312 |
| WorldMirror | **99.99** | **95.81** | **86.28** | 0.096 | 0.058 | 0.490 | **0.010** | **0.009** | **0.297** |

$\pi^3$, with significant improvements of 10.4% and 17.8% in mean accuracy on 7-Scenes and DTU, respectively. Incorporating a single prior can further enhance performance, while the combination of all priors achieves optimal results, which delivers clear gains of 58.1% and 53.1% in mean accuracy on 7-Scenes and NRGBD compared to our no-prior baseline. These results clearly demonstrate our model's ability to effectively leverage prior information for better reconstruction.

**Camera Pose Estimation.** Following the protocol of (Wang et al., 2025c), we test camera pose estimation on three unseen datasets: RealEstate10K (Zhou et al., 2018), Sintel (Bozic et al., 2021), and TUM-dynamics (Sturm et al., 2012). For RealEstate10K, we select 10 fixed images per sequence and examine all pairwise combinations, measuring Relative Rotation Accuracy (RRA), Relative Translation Accuracy (RTA), and Area Under the Curve (AUC) at a 30-degree threshold. For Sintel and TUM-dynamics, we report Absolute Trajectory Error (ATE), Relative Pose Error for translation (RPE trans), and rotation (RPE rot). Tab. 2 demonstrates strong results: our method achieves superior zero-shot performance on RealEstate10K and TUM-dynamics, while maintaining competitive results on Sintel. The performance on Sintel, though slightly below the best methods, is reasonable given the limited outdoor dynamic scenes in our training data.

Table 3: **Surface Normal Estimation on ScanNet, NYUv2, and iBims-1.** We compare with both regression-based and diffusion-based surface normal estimation approaches. EESNU is trained on ScanNet, thus its in-domain performance is omitted.

| Method | ScanNet | | | | NYUv2 | | | | iBims-1 | | | |
|---|---|---|---|---|---|---|---|---|---|---|---|---|
| | mean ↓ | med ↓ | 22.5° ↑ | 30° ↑ | mean ↓ | med ↓ | 22.5° ↑ | 30° ↑ | mean ↓ | med ↓ | 22.5° ↑ | 30° ↑ |
| OASIS (Chen et al., 2020) | 32.8 | 28.5 | 38.5 | 52.6 | 29.2 | 23.4 | 48.4 | 60.7 | 32.6 | 24.6 | 46.6 | 57.4 |
| EESNU (Bae et al., 2021) | - | - | - | - | 16.2 | 8.5 | 77.2 | 83.5 | 20.0 | 8.4 | 73.4 | 78.2 |
| Omnidata v1 (Eftekhar et al., 2021) | 22.9 | 12.3 | 66.1 | 73.2 | 23.1 | 12.9 | 66.3 | 73.6 | 19.0 | 7.5 | 76.1 | 80.1 |
| Omnidata v2 (Kar et al., 2022) | 16.2 | 8.5 | 79.5 | 84.7 | 17.2 | 9.7 | 76.5 | 83.0 | 18.2 | 7.0 | 77.4 | 81.1 |
| DSine (Bae & Davison, 2024) | 16.2 | 8.3 | 78.7 | 84.4 | 16.4 | 8.4 | 77.7 | 83.5 | 17.1 | 6.1 | 79.0 | 82.3 |
| GeoWizard (Fu et al., 2024) | 16.7 | 9.5 | 78.3 | 84.2 | 19.5 | 11.7 | 74.5 | 81.6 | 20.4 | 9.4 | 76.4 | 80.6 |
| StableNormal (Ye et al., 2024b) | 16.0 | 9.9 | 81.5 | 86.5 | 18.5 | 11.2 | 77.5 | 83.6 | 17.9 | 8.5 | 80.4 | 83.9 |
| WorldMirror | **13.8** | **7.3** | **82.5** | **87.3** | 15.1 | **8.0** | 80.1 | 85.7 | 16.6 | 6.4 | 80.1 | 83.7 |

Table 4: **Novel View Synthesis on RealEstate10K and DL3DV.** We compare with feed-forward 3DGS methods under sparse and dense-view settings at the resolution of $518 \times 378$. FLARE focuses on sparse views NVS and thus its performance under dense-view settings is omitted.

| Method | RealEstate10K (2 views) | | | DL3DV (8 views) | | | RealEstate10K (32 views) | | | DL3DV (64 views) | | |
|---|---|---|---|---|---|---|---|---|---|---|---|---|
| | PSNR ↑ | SSIM ↑ | LPIPS ↓ | PSNR ↑ | SSIM ↑ | LPIPS ↓ | PSNR ↑ | SSIM ↑ | LPIPS ↓ | PSNR ↑ | SSIM ↑ | LPIPS ↓ |
| FLARE (Zhang et al., 2025) | 16.33 | 0.574 | 0.410 | 15.35 | 0.516 | 0.591 | - | - | - | - | - | - |
| AnySplat (Jiang et al., 2025) | 17.62 | 0.616 | 0.242 | 18.31 | 0.569 | 0.258 | 19.96 | 0.718 | 0.234 | 18.40 | 0.602 | 0.286 |
| WorldMirror | 20.62 | 0.706 | 0.187 | 20.92 | 0.667 | 0.203 | 25.14 | 0.859 | 0.109 | 21.25 | 0.703 | 0.223 |
| WorldMirror (w/ intrinsics) | 22.03 | 0.765 | 0.165 | 22.08 | 0.723 | 0.175 | 25.71 | 0.877 | **0.101** | 21.55 | 0.731 | 0.207 |
| WorldMirror (w/ camera pose) | 20.84 | 0.713 | 0.182 | 21.18 | 0.674 | 0.197 | 25.14 | 0.865 | 0.107 | 21.28 | 0.700 | 0.222 |
| WorldMirror (w/ intrinsics/camera pose) | **22.30** | **0.774** | **0.155** | **22.15** | **0.726** | **0.174** | **25.77** | **0.879** | **0.101** | **21.66** | **0.736** | **0.204** |

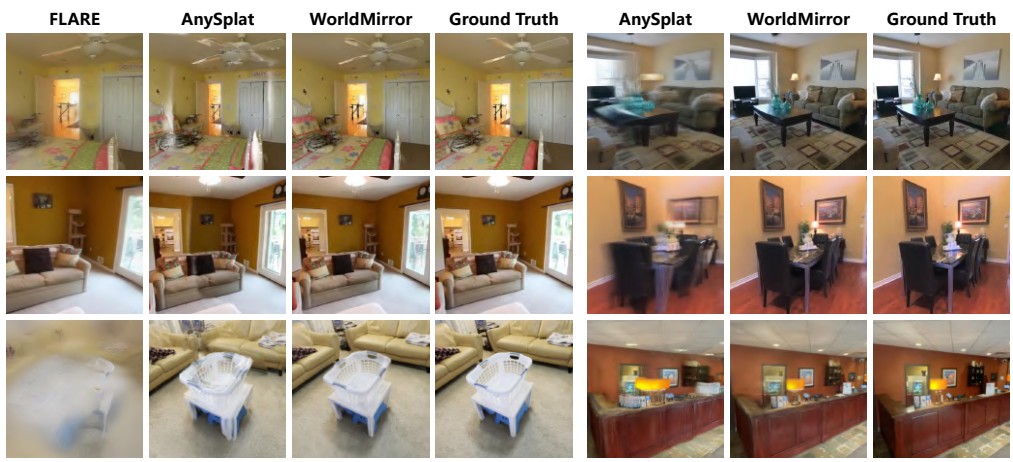

Figure 4: **Qualitative Comparisons of Novel View Synthesis.** We compare with FLARE and AnySplat on RealEstate10K and DL3DV. The first four columns correspond to the sparse-view setting, while the latter three correspond to the dense-view setting. Our approach surpasses baselines in both appearance fidelity and geometric perception.

**Surface Normal Estimation.** Following the protocol from (Bae & Davison, 2024), we evaluate surface normal estimation on three datasets: iBims-1(Koch et al., 2018), NYUv2 (Silberman et al., 2012), and ScanNet (Dai et al., 2017). We measure angular error between predicted and ground truth normal maps, reporting both mean and median errors along with the percentage of pixels below error thresholds of 22.5° and 30.0°. Tab. 3 presents our method's performance across three datasets, demonstrating substantial improvements over existing approaches. The consistent gains across diverse datasets indicate that multi-task frameworks leveraging shared representations can effectively outperform specialized single-task methods.

**Novel View Synthesis.** We evaluate zero-shot novel view synthesis on three datasets: RealEstate10K (Zhou et al., 2018), DL3DV (Ling et al., 2024), and VR-NeRF (Xu et al., 2023) under both sparse-view and dense-view settings. For RealEstate10K, we randomly sample 200 scenes from the NopoSplat (Ye et al., 2024a) test split, using 3 novel views per scene in the sparse-view setting and 4 novel views per scene in the dense-view setting. For DL3DV, we follow the FLARE test

Table 5: Multi-resolution novel view synthesis evaluation on DL3DV under 8, 24, and 64 input views. WorldMirror surpasses SOTA methods (Jiang et al., 2025; Xu et al., 2025b) across resolutions, showing strong generalization to varying input sizes. ∗ denotes using the pose-free optimization (Ye et al., 2024a) for fair comparison with non–pose-free baselines (Xu et al., 2025b).

| Method | Prior Condition | Resolution | DL3DV (8 Views) | | | DL3DV (24 Views) | | | DL3DV (64 Views) | | |
|---|---|---|---|---|---|---|---|---|---|---|---|
| | | | PSNR ↑ | SSIM ↑ | LPIPS ↓ | PSNR ↑ | SSIM ↑ | LPIPS ↓ | PSNR ↑ | SSIM ↑ | LPIPS ↓ |
| AnySplat | None | 448×252 | 15.62 | 0.453 | 0.305 | 18.68 | 0.571 | **0.221** | 19.50 | 0.610 | **0.208** |
| WorldMirror | None | 448×252 | **17.50** | **0.518** | **0.303** | **19.15** | **0.602** | 0.241 | **19.51** | **0.626** | 0.239 |
| FLARE | Intrinsics | 256×256 | 14.77 | 0.412 | 0.647 | 14.11 | 0.398 | 0.761 | - | - | - |
| WorldMirror | Intrinsics | 252×252 | **16.83** | **0.480** | **0.320** | **18.76** | **0.574** | **0.244** | 19.10 | 0.593 | 0.240 |
| DepthSplat | Intrinsics + Poses | 448×256 | 18.79 | 0.619 | 0.316 | 18.71 | 0.643 | 0.313 | 16.80 | 0.551 | 0.416 |
| WorldMirror* | Intrinsics + Poses | 448×252 | **19.08** | **0.624** | **0.261** | **20.24** | **0.675** | **0.221** | **20.30** | **0.680** | **0.226** |

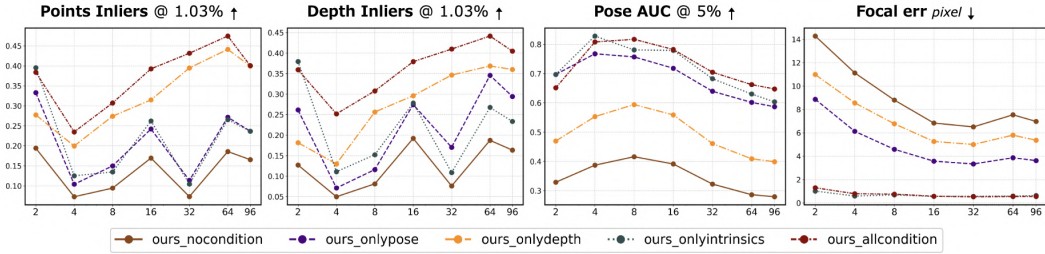

Figure 5: **Geometric Priors Boosts Model's Feed-Forward Performance across All Tasks.** Incorporating a single modality not only enhances predictions for its corresponding task but also improves performance across other tasks. This suggests that modal information enables the model to develop a more comprehensive understanding of the overall geometry.

split and evaluate in 112 unseen scenes, each containing 9 novel views. For VR-NeRF, consistent with AnySplat, we select 5 scenes, each with 64 input views and 6 novel views. For calculating the rendering metrics, we follow the *test-time camera pose alignment* introduced by AnySplat to ensure fair evaluation. Tab. 4 reports the quantitative evaluation results at a unified resolution of $518 \times 378$ for novel view synthesis under the feed-forward setting. Our method achieves substantial improvements over the previous state-of-the-art AnySplat, with consistent gains across all metrics on both datasets, demonstrating the effectiveness of our unified geometric representation for high-quality view synthesis.

To address differences in input resolutions across prior NVS methods, we include a multi-resolution comparison with recent SOTA approaches (Jiang et al., 2025; Xu et al., 2025b) on DL3DV. As shown in Table 5, our method consistently outperforms DepthSplat at 8, 24, and 64 views, with gains increasing as the number of input views grows. It is worth noting that our model is trained with dynamic input resolutions, enabling it to handle varying resolutions during inference. As demonstrated in the table, our approach generalizes robustly across a wide range of resolutions and consistently outperforms the baselines.

## 4.2 EVALUATION ON DIFFERENT INPUT CONFIGURATIONS

To demonstrate the benefits of incorporating priors into model predictions, we evaluate model performance across various input configurations. We present four key metrics: the inlier ratio at a relative threshold of 1.03% of points and depths, the area under the curve at a 5° error threshold (AUC@5), and the average focal error in pixels, measured across the ETH3D (Schops et al., 2017) and DTU (Jensen et al., 2014) datasets. As shown in Fig.5, incorporating even a single modality prior yields dual benefits: it enhances both the corresponding task prediction and the model's capacity to infer other geometric attributes. Fig.6 illustrates how different priors contribute to reconstruction quality. Camera poses enable the model to capture global scene geometry, calibrated intrinsics resolve scale ambiguity, while depth priors offer pixel-level constraints that prove particularly valuable for reconstructing geometrically complex regions. These findings confirm that multi-modal priors work synergistically, where each modality provides complementary geometric constraints that collectively improve the model's understanding of 3D scene structure.

Table 6: Comparison with Pow3R and MapAnything under different prior conditions on 7-Scenes, NRGBD, and DTU datasets. Pow3R (pro) refers to Best results are in **bold**, second best are underlined.

| Method | Prior Condition | 7S-Acc. ↓ | 7S-Comp. ↓ | NRGBD-Acc. ↓ | NRGBD-Comp. ↓ | DTU-Acc. ↓ | DTU-Comp. ↓ |
|---|---|---|---|---|---|---|---|
| Pow3R (pro) | None | 0.103 | 0.174 | 0.121 | 0.099 | 5.104 | 2.863 |
| MapAnything | | 0.075 | 0.093 | 0.088 | 0.100 | 1.997 | 2.068 |
| WorldMirror | | **0.044** | **0.050** | **0.038** | **0.041** | **0.982** | **1.486** |
| Pow3R (pro) | Intrinsics | 0.104 | 0.175 | 0.120 | 0.101 | 4.336 | 2.711 |
| MapAnything | | 0.073 | 0.090 | 0.086 | 0.100 | 2.309 | 2.011 |
| WorldMirror | | **0.048** | **0.051** | **0.038** | **0.042** | **0.948** | **1.579** |
| Pow3R (pro) | Depth | 0.103 | 0.174 | 0.121 | 0.099 | 5.104 | 2.863 |
| MapAnything | | 0.067 | 0.080 | 0.065 | 0.073 | 3.879 | 2.403 |
| WorldMirror | | **0.058** | **0.060** | **0.028** | **0.027** | **0.790** | **0.977** |
| Pow3R (pro) | Camera Pose | 0.049 | 0.049 | 0.074 | 0.062 | 4.342 | 2.465 |
| MapAnything | | 0.029 | **0.032** | 0.047 | 0.045 | 2.394 | 2.073 |
| WorldMirror | | **0.023** | 0.035 | **0.023** | **0.026** | **0.967** | **1.502** |
| Pow3R (pro) | All Priors | 0.049 | 0.046 | 0.072 | 0.060 | 3.526 | 2.309 |
| MapAnything | | **0.012** | **0.013** | 0.016 | 0.013 | 1.707 | **0.989** |
| WorldMirror | | 0.018 | 0.024 | **0.013** | **0.011** | **0.717** | 0.947 |

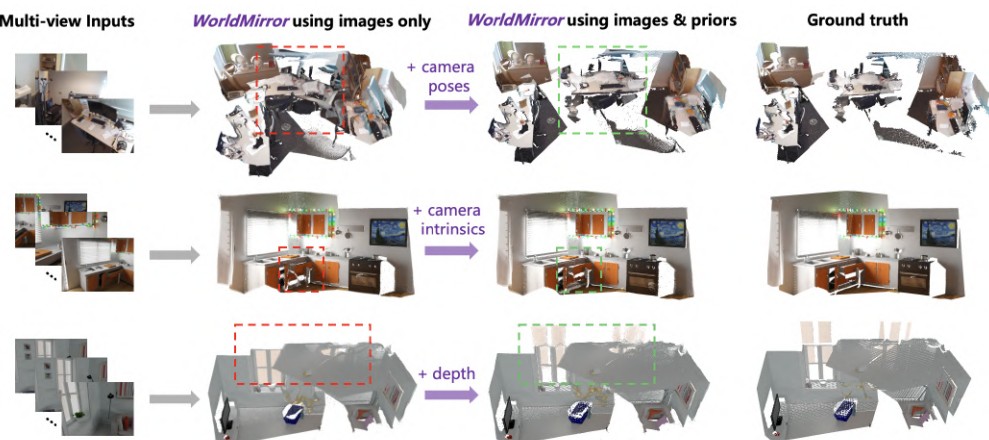

Figure 6: **Geometric Priors Unlock Enhanced Scene Reconstruction of *WorldMirror*.** (Top) Camera poses help the model to capture relative view positions accurately. (Middle) Calibrated intrinsic enhances the reconstruction by enabling precise projection modeling and geometry alignment. (Bottom) Depth guidance enables the network to better handle challenging reconstruction scenarios, like perspective distortion, unusual geometric configurations, or partial occlusions.

## 4.3 COMPARISON WITH PRIOR-GUIDED METHODS

We also provide comprehensive comparisons with recent prior-guided methods Pow3R (Jang et al., 2025) and MapAnything (Keetha et al., 2025) under different prior conditions. Note that Pow3R was originally designed for 2-view reconstruction; to extend it to multi-view scenarios, we incorporate Procrustes alignment for camera pose estimation from the predicted point clouds. The results are organized by prior availability in Table 6.

The results demonstrate that *WorldMirror* consistently outperforms both Pow3R and MapAnything across most prior conditions and metrics. Compared to Pow3R, our method adopts cleaner and more multi-view-friendly embedding strategies for calibrated intrinsics and camera poses, which better preserves geometric consistency across multiple views. Compared to MapAnything, our model benefits from fine-tuning on VGGT, leading to better generalization on out-of-domain data. These architectural and training improvements enable *WorldMirror* to achieve superior reconstruction quality across diverse scenarios.

## 4.4 ABLATION STUDY

**Prior Embedding Ablation.** We explore different ways of embedding priors in Tab. 7. For camera poses, we experiment with (1) dense Plücker ray embeddings that are added element-wise to the

Table 7: **Prior Embedding Ablation.** Results are averaged over ETH3D and DTU datasets with 10 views as input. 'Single token' offers both superior performance and high efficiency.

| Prior embedding | Extra Params | Focal acc@1.03↑ | Depth $\tau$@1.03↑ | RRA@5↑ | Pose RTA@5↑ | AUC@5↑ | Point $\tau$@1.03↑ | Avg. ↑ |
|---|---|---|---|---|---|---|---|---|
| **Input: images & poses** | | | | | | | | |
| Dense Plücker | 9.02M | 33.07 | 31.00 | 98.59 | 93.52 | 72.74 | 33.74 | 60.44 |
| Single Token | 1.06M | 33.82 | 28.02 | 98.89 | 92.57 | 74.55 | 38.51 | 61.06 |
| **Input: images & intrinsics** | | | | | | | | |
| Dense Raymap | 6.65M | 86.48 | 29.36 | 97.17 | 88.48 | 60.57 | 37.40 | 66.58 |
| Single Token | 1.06M | 84.43 | 34.70 | 98.18 | 93.64 | 66.52 | 36.29 | 68.96 |

Table 8: **Novel View Synthsis Ablation.** Results are from RealEstate10K, DL3DV, and VR-NeRF.

| Method | RealEstate10K (2 views) | | | DL3DV (8 views) | | | VR-NeRF (32 views) | | |
|---|---|---|---|---|---|---|---|---|---|
| | PSNR ↑ | SSIM ↑ | LPIPS ↓ | PSNR ↑ | SSIM ↑ | LPIPS ↓ | PSNR ↑ | SSIM ↑ | LPIPS ↓ |
| w/o GT Cameras | **20.30** | 0.691 | 0.193 | 20.69 | 0.666 | 0.206 | 24.76 | 0.788 | 0.197 |
| w/o Novel Views | 18.51 | 0.651 | 0.215 | 20.21 | 0.664 | **0.196** | 24.35 | 0.781 | 0.199 |
| w/o GS DPT | 20.28 | 0.693 | 0.195 | 20.55 | 0.667 | 0.218 | 25.08 | 0.798 | **0.191** |
| Ours | 20.29 | **0.693** | **0.192** | **20.91** | **0.671** | 0.198 | **25.75** | **0.811** | 0.198 |

image tokens, and (2) a single token concatenation approach where the pose is compressed into a single token and concatenated to the sequence. For camera intrinsics, we similarly compare dense raymap embeddings that are added to the image tokens versus a single token. Our experiments reveal that the single token approach achieves better performance for embedding both camera poses and intrinsics, suggesting that a compact global representation is more effective than dense per-pixel conditioning while being more efficient.

**Novel View Synthesis Ablation.** Tab. 8 reports ablation analysis on the novel view synthesis: (1) To examine the importance of using ground-truth camera parameters for novel view rendering, we replace the ground-truth poses and intrinsic matrices in our method with those predicted by the camera head for computing 3DGS positions and rendering. (2) To assess the necessity of supervising 3DGS rendering not only on input views but also on novel views, we perform an ablation similar to (Jiang et al., 2025), where no novel-view rendering loss is applied. (3) The GS head predicts all Gaussian attributes except positions, while the positions are derived from the depth maps estimated by the Depth head. These studies confirm that both our 3DGS prediction framework and training strategy are crucial, and removing any component degrades novel view rendering performance.

## 5 CONCLUSION

We presented *WorldMirror*, a unified feed-forward model that addresses versatile 3D reconstruction tasks. By flexibly incorporating diverse geometric priors and generating multiple 3D representations simultaneously, our framework demonstrates that a single model can effectively handle various 3D reconstruction tasks without task-specific specialization. *WorldMirror* achieves state-of-the-art performance across dense reconstruction, multi-view depth estimation, surface normal prediction, and novel view synthesis, while maintaining feed-forward efficiency. The model's ability to leverage available priors enables robust reconstruction in challenging scenarios, and its multi-task design ensures geometric consistency across different outputs. Our work shows that unified, prior-aware architectures offer a promising direction for comprehensive and efficient 3D scene understanding.

## ETHICS STATEMENT

*WorldMirror* enables efficient 3D scene reconstruction across multiple applications, making advanced geometric prediction more accessible. While benefiting researchers and small teams, we recognize concerns about privacy implications, potential misrepresentation of environments, and algorithmic biases. We encourage ongoing research into verification methods and ethical guidelines for responsible implementation of 3D reconstruction technologies.

## REPRODUCIBILITY STATEMENT

We commit to full transparency by releasing our codebase, model weights, and implementation scripts upon publication. Appendix A details our architecture, training protocols, and hyperparameters. All evaluation metrics and protocols are clearly specified, along with hardware requirements to ensure complete reproducibility of our results.

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

# A  MODEL TRAINING

## A.1  TRAINING LOSSES

Our model is trained end-to-end by minimizing a composite loss function, $\mathcal{L}$, which integrates supervision for all prediction tasks:

$$\mathcal{L} = \lambda_{\text{points}}\mathcal{L}_{\text{points}} + \lambda_{\text{depth}}\mathcal{L}_{\text{depth}} + \lambda_{\text{cam}}\mathcal{L}_{\text{cam}} + \lambda_{\text{normal}}\mathcal{L}_{\text{normal}} + \lambda_{\text{3dgs}}\mathcal{L}_{\text{3dgs}}. \tag{6}$$

We follow VGGT to implement $\mathcal{L}_{\text{cam}}$, $\mathcal{L}_{\text{points}}$, and $\mathcal{L}_{\text{depth}}$. Specifically, we use a gradient-based term to supervise the predicted point $\hat{P}_i$:

$$\mathcal{L}_{\text{point}} = \sum_{i=1}^{N} \|\Sigma_i^P \odot (\hat{\boldsymbol{P}}_i - \boldsymbol{P}_i)\| + \|\Sigma_i^P \odot (\nabla\hat{\boldsymbol{P}}_i - \nabla\boldsymbol{P}_i)\| - \alpha \log \Sigma_i^P, \tag{7}$$

where $\odot$ is the channel-broadcast element-wise product and $\Sigma_i^P$ refers to the point uncertainty. The depth loss $\mathcal{L}_{\text{depth}}$ is analogous to $\mathcal{L}_{\text{point}}$ but replaces the point with depth. For camera loss $\mathcal{L}_{\text{cam}}$, we implement a Huber loss $\|\cdot\|_\epsilon$ to supervise the predicted camera $\boldsymbol{E}_i$:

$$\mathcal{L}_{\text{cam}} = \Sigma_{i=1}^{N}\|\boldsymbol{E}_i - \hat{\boldsymbol{E}}_i\|_\epsilon. \tag{8}$$

To supervise the predicted surface normals $\hat{\boldsymbol{E}}_i$, we use Angle Loss (AL), which effectively measures the directional deviation between predicted and ground truth normal vectors. The normal loss function is specifically defined as:

$$\mathcal{L}_{\text{normal}} = \Sigma_{i=1}^{N}\alpha_l \cdot (1 - |\hat{\boldsymbol{N}}_i \cdot \boldsymbol{N}_i|). \tag{9}$$

To enhance robustness in novel views, at each training iteration, we partition the input views $I$ into $K$ candidate context and novel view splits. The pixel overlap rate between the ground truth depth map and camera parameters is computed for each novel view in the context of the candidate context views. The split with the highest pixel overlap rate is selected, with the corresponding context views and novel views being used for further training. Next, based on the selected context images, we regress the 3DGS positions and properties, and render both context view images and novel view images $\hat{I}$. Then, the RGB rendering loss across all views is defined as follows:

$$\mathcal{L}_{rgb} = \Sigma_{i=1}^{N}\|I_i[M_i] - \hat{I}_i[M_i]\| + \lambda_{\text{lpips}}\text{LPIPS}(I_i[M_i], \hat{I}_i[M_i]), \tag{10}$$

where $M$ denotes the mask indicating whether the pixels in the current view are visible from the context views, analogous to the novel view mask introduced in Smart et al. (2024).

To explicitly supervise the locations of the 3D Gaussian splats, we introduce the depth supervision loss $\mathcal{L}_{\text{gsdepth}}$, which enforces consistency between the ground truth depth map and the depth map predicted by the GS head. The formulation of $\mathcal{L}_{gsdepth}$ follows the same definition as Eq. 7. It is worth noting that, instead of using the depth estimated by the depth head to compute the Gaussian positions, we rely on the GS head to directly predict both the positions and other attributes of the splats. This design choice is further validated in our ablation studies (see Tab. 8). However, due to inherent ambiguities in multi-view rendering and potential noise in the ground truth depth, relying solely on $\mathcal{L}_{\text{rgb}}$ and $\mathcal{L}_{gsdepth}$ often leads to the presence of floating points in the predicted 3DGS. To mitigate this issue, we introduce a gradient consistency loss $\mathcal{L}_{\text{consis}}$, which regularizes the gradients of the GS-rendered depth map $\tilde{D}$ to be consistent with the pseudo depth $\hat{D}$ predicted by the depth head:

$$\mathcal{L}_{\text{consis}} = \Sigma_{i=1}^{N}\|\nabla\hat{D}_i[\hat{M}_i] - \nabla\tilde{D}_i[\hat{M}_i]\|, \tag{11}$$

where $\hat{M}$ is the depth confidence mask corresponding to the top $30\%$-quantile of the confidence map. Finally, the 3DGS loss is defined as $\mathcal{L}_{\text{3dgs}} = \mathcal{L}_{\text{rgb}} + \lambda_{\text{gsdepth}}\mathcal{L}_{\text{gsdepth}} + \lambda_{\text{consis}}\mathcal{L}_{\text{consis}}$.

## A.2  TRAINING SETTINGS

**Implementation Details.** Our model undergoes a two-phase training process. Initially, we train for 100 epochs using multi-modal prior prompting with a normal head, followed by 50 epochs of fine-tuning with a Gaussian head. Throughout both phases, we implement dynamic image resolutions,

Table 9: **Monocular and Video Depth Estimation on NYUv2, Sintel, and KITTI.**

| Method | NYU-v2 (Monocular) | | Sintel (Monocular) | | KITTI (Video) | | Sintel (Video) | |
|---|---|---|---|---|---|---|---|---|
| | Abs Rel ↓ | $\delta < 1.25$ ↑ | Abs Rel ↓ | $\delta < 1.25$ ↑ | Abs Rel ↓ | $\delta < 1.25$ ↑ | Abs Rel ↓ | $\delta < 1.25$ ↑ |
| DUSt3R (Wang et al., 2024) | 0.081 | 0.909 | 0.488 | 0.532 | 0.143 | 0.814 | 0.662 | 0.434 |
| MASt3R (Leroy et al., 2024) | 0.11 | 0.865 | 0.413 | 0.569 | 0.115 | 0.848 | 0.558 | 0.487 |
| MonST3R (Zhang et al., 2024) | 0.094 | 0.887 | 0.492 | 0.525 | 0.107 | 0.884 | 0.399 | 0.519 |
| Fast3R (Yang et al., 2025) | 0.093 | 0.898 | 0.544 | 0.509 | 0.138 | 0.834 | 0.638 | 0.422 |
| CUT3R (Wang et al., 2025b) | 0.081 | 0.914 | 0.418 | 0.52 | 0.122 | 0.876 | 0.417 | 0.507 |
| FLARE (Zhang et al., 2025) | 0.089 | 0.898 | 0.606 | 0.402 | 0.356 | 0.57 | 0.729 | 0.336 |
| VGGT (Wang et al., 2025a) | 0.056 | 0.951 | 0.606 | 0.599 | 0.062 | 0.969 | 0.299 | 0.638 |
| $\pi^3$ (Wang et al., 2025c) | 0.054 | 0.956 | **0.277** | 0.614 | **0.038** | **0.986** | **0.233** | 0.664 |
| **WorldMirror** | **0.052** | **0.957** | 0.339 | **0.624** | 0.063 | 0.968 | 0.289 | **0.668** |

Table 10: **Novel View Synthesis with 3DGS Optimization on RealEsate10K, DL3DV, and VRN-eRF.** In Post-Optimization, the *random point cloud* refers to initializing Gaussian positions randomly, whereas the *predicted point cloud* uses the point cloud estimated by our method as the initialization of Gaussian positions.

| Method | Iterations | RealEstate10K (32 views) | | | | DL3DV (64 views) | | | | VRNeRF (64 views) | | | |
|---|---|---|---|---|---|---|---|---|---|---|---|---|---|
| | | PSNR ↑ | SSIM ↑ | LPIPS ↓ | Time ↓ | PSNR ↑ | SSIM ↑ | LPIPS ↓ | Time ↓ | PSNR ↑ | SSIM ↑ | LPIPS ↓ | Time ↓ |
| **Feedforward** | | | | | | | | | | | | | |
| AnySplat | - | 19.96 | 0.718 | 0.234 | <2s | 18.40 | 0.602 | 0.286 | <2s | 22.11 | 0.759 | 0.288 | <2s |
| WorldMirror | - | 25.14 | 0.859 | 0.109 | <2s | 21.25 | 0.703 | 0.223 | <2s | 25.77 | 0.830 | **0.208** | <2s |
| **Post Optimization** | | | | | | | | | | | | | |
| *random points cloud* | 3,000 | 26.03 | 0.875 | 0.145 | 19s | 23.61 | 0.765 | 0.244 | 21s | **26.45** | 0.840 | 0.259 | 21s |
| *predicted points cloud* | 1,000 | 27.29 | 0.906 | 0.092 | 10s | 23.43 | 0.772 | 0.248 | 12s | 25.19 | 0.841 | 0.257 | 11s |
| AnySplat | 1,000 | 23.85 | 0.834 | 0.192 | 23s | 20.84 | 0.695 | 0.287 | 55s | 23.19 | 0.782 | 0.322 | 33s |
| AnySplat | 3,000 | 26.03 | 0.870 | 0.155 | 56s | 22.20 | 0.723 | 0.226 | 126s | 24.64 | 0.798 | 0.272 | 65s |
| WorldMirror | 1,000 | **27.79** | **0.915** | **0.076** | 23s | **23.86** | **0.786** | **0.172** | 45s | 25.98 | **0.845** | 0.214 | 38s |

maintaining total pixel counts between 100,000 and 250,000, while sampling aspect ratios from 0.5 to 2.0. We employ a dynamic batch sizing approach similar to VGGT, processing 24 images per GPU across a cluster of 32 H20 GPUs. Our optimization strategy features parameter-specific learning rates: 2e-5 for patch embedding layers, 1e-4 for alternated attention modules and pre-trained pointmap, depth, and camera head, and 2e-4 for newly introduced parameters. We use a CosineAnnealing scheduler that gradually decreases from maximum to minimum values following a cosine curve. For our composite loss function, we carefully balance component weights as follows: $\lambda_{\text{points}} = 1.0$, $\lambda_{\text{depth}} = 1.0$, $\lambda_{\text{cam}} = 5.0$, $\lambda_{\text{normal}} = 1.0$, $\lambda_{\text{3dgs}} = 1.0$, $\lambda_{\text{lpips}} = 0.05$, $\lambda_{\text{gsdepth}} = 0.1$, $\lambda_{\text{consis}} = 0.1$.

**Dynamic Prior Injection Scheme.** Specifically, we randomly toggle each prior modality with a probability of 0.5 during training. When a particular prior is disabled, we set the corresponding tokens to zero. This straightforward approach offers several advantages: it enhances model robustness by forcing the network to handle missing information, enables graceful degradation when certain priors are unavailable during inference, and creates a single unified model capable of operating across different prior combinations.

**Curriculum Learning Strategy.** During training, we employ a systematic curriculum learning strategy designed to optimize training efficiency and enhance performance by progressing from simple to complex across task sequencing, data scheduling, and resolution.

For task sequencing, initially, we jointly train the multi-modal prior prompting module with other parameters initialized from the pretrained weights of VGGT, which establishes a foundational capability of prior-aware prediction. We then incorporate the normal prediction task into the joint training scheme. Finally, we freeze all model parameters and exclusively train the 3DGS head for 3DGS attributes prediction. This progressive task sequencing strategy ensures effective training for universal geometric prediction with any prior combination.

For data scheduling, we equip the initial training phase with a comprehensive dataset of both real and synthetic data, which exposes the model to a diverse data distribution for improving the generalization capabilities and preventing overfitting. Following this, the model undergoes a fine-tuning stage using only synthetic data with high-quality annotations of camera, depth, and surface normal, which mitigates the impact of annotation noise inherent in real-world datasets, guiding the model to learn more precise and reliable patterns.

Table 11: Novel view synthesis results on MatrixCity (Li et al., 2023) using 100, 150, and 200 input views. *WorldMirror* consistently outperforms prior feed-forward and optimization-based methods across most metrics.

| Method | MatrixCity (100 views) | | | MatrixCity (150 views) | | | MatrixCity (200 views) | | |
|---|---|---|---|---|---|---|---|---|---|
| | PSNR ↑ | SSIM ↑ | LPIPS ↓ | PSNR ↑ | SSIM ↑ | LPIPS ↓ | PSNR ↑ | SSIM ↑ | LPIPS ↓ |
| 3D-GS (Kerbl et al., 2023) | 18.21 | 0.568 | 0.445 | 18.86 | 0.593 | 0.412 | 19.54 | 0.612 | 0.388 |
| Mip-Splatting (Yu et al., 2024) | 17.97 | 0.536 | 0.450 | 18.24 | 0.579 | 0.438 | 18.63 | 0.588 | 0.414 |
| AnySplat (Jiang et al., 2025) | 20.51 | 0.620 | **0.347** | 19.24 | 0.601 | 0.399 | 19.18 | 0.605 | 0.397 |
| **WorldMirror** | **20.88** | **0.640** | 0.360 | **20.62** | **0.626** | **0.370** | **20.36** | **0.630** | **0.375** |

Table 12: Two-view NVS comparison on RealEstate10K and DL3DV. WorldMirror demonstrates strong generalization ability, even without being trained specifically for the two-view NVS setting.

| Method | Prior-Type | RealEstate10K (2 views) | | | DL3DV (2 views) | | |
|---|---|---|---|---|---|---|---|
| | | PSNR ↑ | SSIM ↑ | LPIPS ↓ | PSNR ↑ | SSIM ↑ | LPIPS ↓ |
| NoPoSplat (Ye et al., 2024a) | Intrinsics | **25.06** | **0.836** | 0.164 | 19.00 | 0.575 | 0.350 |
| AnySplat (Jiang et al., 2025) | None | 18.01 | 0.602 | 0.207 | 13.56 | 0.368 | 0.338 |
| WorldMirror | None | 23.48 | 0.805 | 0.124 | 18.41 | 0.582 | 0.270 |
| WorldMirror | Intrinsics | 23.89 | 0.826 | **0.113** | **19.08** | **0.636** | **0.250** |

For training resolution, we use a progressive resolution warm-up, beginning with low-resolution inputs and outputs to ensure stable and rapid initial convergence, then gradually increasing the resolution to enhance the model's ability to perceive fine details.

**Training Data.** The training data comprises a diverse collection of 15 datasets spanning various scene types and capture conditions. This heterogeneous mix includes both established benchmarks and recent collections: DL3DV (Ling et al., 2024), BlenderMVS (Yao et al., 2020), TartanAir (Wang et al., 2020), ASE (Pan et al., 2023), Unreal4K (Tosi et al., 2021), Habitat (Savva et al., 2019), MapFree (Arnold et al., 2022), MVS-Synth (Huang et al., 2018), ArkitScenes (Baruch et al., 2021), ScanNet++ (Yeshwanth et al., 2023), MegaDepth (Li & Snavely, 2018), Hypersim (Roberts et al., 2021), Matterport3D (Chang et al., 2017), Co3dv2 (Reizenstein et al., 2021), and WildRGBD (Xia et al., 2024) datasets. This extensive dataset aggregation provides rich supervision across indoor/outdoor environments, real/synthetic scenes, and static/dynamic objects, enabling our model to learn generalizable geometric representations.

# B ADDITIONAL COMPARISONS

## B.1 MONOCULAR AND VIDEO DEPTH BENCHMARK

In Table 9, we evaluate *WorldMirror* in comparison with contemporary approaches for both single-view and sequential depth estimation across diverse input scenarios. Despite *WorldMirror* not being explicitly optimized for monocular metric depth inference, it delivers performance that matches or exceeds current leading methods. When processing video sequences, *WorldMirror* produces results that rival specialized feed-forward reconstruction frameworks. We note a modest performance gap on the KITTI benchmark relative to $\pi^3$, which we attribute to the under-representation of urban driving environments in our training distribution. Future iterations of our work will incorporate a more comprehensive collection of street-level imagery to enhance generalization to such scenarios.

## B.2 NOVEL VIEW SYNTHESIS WITH OPTIMIZATION

Although recent feed-forward pipelines are capable of synthesizing competitive 3D Gaussian splats (3DGS) within seconds, they inevitably suffer from errors introduced by single-pass predictions, such as suboptimal Gaussian placement and appearance. We hypothesize that incorporating a brief post-optimization stage—initialized with either our predicted point cloud or 3DGS primitives—can significantly refine both geometry and appearance at only modest additional cost, thereby accelerating the convergence of 3DGS training and enhancing rendering quality.

As shown in Tab. 10, we compare (i) feed-forward baselines and (ii) post-optimization with 3,000 or 1,000 iterations, initialized either from a random point cloud or from feed-forward 3DGS primitives. The camera parameters for optimizing 3DGS are obtained from the feed-forward outputs of the chosen method. Our predicted point cloud, camera, and 3DGS primitives provide a robust and high-quality initialization for 3DGS optimization, significantly accelerating the training process and consistently surpassing baseline methods across all metrics.

### B.3 NOVEL VIEW SYNTHESIS WITH LARGE-SCALE MULTI-VIEW INPUTS

We evaluate novel view synthesis with 100, 150, and 200 input views on the MatrixCity dataset (Li et al., 2023). Following AnySplat (Jiang et al., 2025), we use a resolution of $448 \times 448$ for all methods and compare *WorldMirror* with AnySplat (Jiang et al., 2025), 3D-GS (Kerbl et al., 2023), and Mip-Splatting (Yu et al., 2024). As shown in the Table 11, the experiment demonstrates that our method generalizes far beyond the maximum of 24 input views used during training, and achieves superior performance to both feed-forward and optimization-based approaches, without any post-processing or additional refinement.

### B.4 NOVEL VIEW SYNTHESIS IN THE TWO-VIEW SETTING

We conduct two-view NVS experiments on both RealEstate10K (Zhou et al., 2018) and DL3DV (Ling et al., 2024) compared with NoPoSplat (Ye et al., 2024a). NoPoSplat takes inputs at a resolution of $256 \times 256$, whereas AnySplat (Jiang et al., 2025) and *WorldMirror* use input images at $406 \times 406$ (center-cropping and resizing to ensure an equivalent receptive field to $256 \times 256$). All methods are evaluated at $256 \times 256$ by downsampling the rendered results. As shown in the Table 12, our model is not trained specifically on RealEstate10K for the two-view setting, yet it achieves performance comparable to NoPoSplat across most metrics. Moreover, when compared to AnySplat, which more closely matches our training configuration, our method substantially outperforms it.

Table 13: Robustness evaluation of WorldMirror with noisy priors on 7-Scenes and DTU datasets. The model exhibits graceful degradation under various noise conditions.

| Prior Type | Noise Level | 7S-Acc. ↓ | 7S-Comp. ↓ | DTU-Acc. ↓ | DTU-Comp. ↓ |
|---|---|---|---|---|---|
| None (baseline) | - | 0.044 | 0.050 | 0.982 | 1.486 |
| Pose (Rotation) | 0° (clean) | **0.022** | **0.035** | **0.966** | 1.502 |
| | 20° | 0.023 | **0.035** | 0.976 | 1.483 |
| | 40° | 0.025 | 0.036 | 1.017 | **1.466** |
| Pose (Translation) | $\sigma = 0.0$ (clean) | **0.022** | **0.035** | **0.966** | 1.502 |
| | $\sigma = 0.05$ | **0.022** | **0.035** | 0.967 | 1.503 |
| | $\sigma = 0.1$ | 0.024 | 0.036 | 0.967 | 1.503 |
| Intrinsics | 1.0× (clean) | 0.047 | 0.051 | **0.948** | **1.579** |
| | 0.8× | **0.044** | **0.049** | 1.047 | 1.740 |
| | 0.6× | 0.051 | 0.054 | 2.691 | 1.830 |
| Depth | $\sigma = 0.0$ (clean) | **0.058** | **0.060** | **0.790** | **0.977** |
| | $\sigma = 0.05$ | **0.058** | 0.061 | 0.795 | 0.980 |
| | $\sigma = 0.1$ | 0.064 | 0.071 | 1.387 | 1.820 |

## C ROBUSTNESS TO NOISY OR LOW-QUALITY PRIORS

To evaluate the robustness of our method to noisy or low-quality priors, we conducted comprehensive experiments with controlled noise injection across different prior types. Following Pow3R (Jang et al., 2025), we designed realistic noise patterns that simulate real-world sensor inaccuracies and calibration errors.

**Noise injection settings.** For Camera Pose, we follow Pow3R and apply rotational noise by rotating the rotation matrix clockwise by 0° (clean), 20°, and 40°. In addition to rotation errors, we also consider isotropic Gaussian noise on the camera translation component with standard deviation $\sigma = 0.0$ (clean), 0.05, and 0.1. For Camera Intrinsics, we follow Pow3R and scale the intrinsic parameters

by factors of $1.0\times$ (clean), $0.8\times$, and $0.6\times$. For Depth Prior, we apply pixel-wise multiplicative noise to depth maps by scaling each pixel with a random factor $\sim \mathcal{N}(1.0, \sigma)$, where $\sigma = 0.0$ (clean), 0.05, and 0.1. We evaluate reconstruction quality on 7-Scenes and DTU datasets, comparing against the baseline model without any priors (None). The results are shown in Table 13.

Our model exhibits graceful performance degradation as noise increases, indicating robust feature learning. Even with moderate noise (20° rotation, $0.8\times$ intrinsics, $\sigma = 0.05$ depth), noisy priors still provide meaningful guidance compared to the no-prior baseline in many cases. Camera pose priors show strong robustness up to 20° rotation error, maintaining better performance than the baseline. Depth priors are more sensitive to noise due to direct geometric guidance, but still maintain reasonable performance with $\sigma = 0.05$. These results demonstrate that *WorldMirror* can effectively leverage imperfect priors while maintaining robustness to various noise levels.

# D    MORE ANALYSIS OF DEPTH PRIOR DESIGN

## D.1    DEPTH PRIOR INJECTION: ADDITION VS. CONCATENATION

We compared two depth prior injection strategies to validate our design choice: (1) Concat: Concatenate depth features with image tokens along the token dimension; (2) Addition (Ours): Directly add depth features to image tokens. The quantitative comparison is presented in Table 14.

Table 14: Comparison of depth prior injection strategies on 7-Scenes and DTU datasets. Inference time and FLOPs are measured for depth prior injection on a single image with resolution $518 \times 378$.

| Depth Input | 7S-Acc. ↓ | 7S-Comp. ↓ | DTU-Acc. ↓ | DTU-Comp. ↓ | Inference Time (s) ↓ | TFLOPs ↓ |
|---|---|---|---|---|---|---|
| Concat | 0.06 | 0.07 | 0.82 | 0.98 | 0.11 | 1.74 |
| Addition (Ours) | **0.06** | **0.06** | **0.79** | **0.97** | **0.09** | **1.14** |

Addition achieves superior accuracy with optimal efficiency, while concatenation increases computational cost by 52.6% with comparable performance. Token addition offers two advantages: (1) Enhanced Spatial Alignment: Direct addition maintains pixel-wise correspondence between depth and image features, enabling more effective feature integration. (2) Computational Efficiency: Additive fusion introduces minimal overhead, avoiding expensive attention operations that would increase computational costs.

## D.2    DEPTH NORMALIZATION AND SCALE INFORMATION

To demonstrate that depth prior normalization improves relative geometry, we conduct an ablation study comparing normalized vs. unnormalized depth inputs on 7-Scenes and DTU datasets, as shown in Table 15.

Table 15: Comparison of normalized vs. unnormalized depth inputs on 7-Scenes and DTU datasets. Normalized depth provides more stable geometric priors for relative-scale reconstruction.

| Depth Input | 7S-Acc. ↓ | 7S-Comp. ↓ | 7S-NC1 ↑ | 7S-NC2 ↑ | DTU-Acc. ↓ | DTU-Comp. ↓ | DTU-NC1 ↑ | DTU-NC2 ↑ |
|---|---|---|---|---|---|---|---|---|
| Unnormalized | 0.11 | 0.12 | 0.66 | 0.67 | 2.60 | 5.96 | 0.52 | 0.51 |
| Normalized (Ours) | **0.06** | **0.06** | **0.77** | **0.78** | **0.79** | **0.97** | **0.70** | **0.71** |

The substantial improvement demonstrates that normalized depth effectively provides geometric priors for better relative-scale reconstruction. We hypothesize this is because normalization offers several key advantages: (1) Consistent Feature Range: Normalization maps depth values from diverse scenes with varying absolute scales into a unified [0,1] range, enabling the model to learn consistent depth-to-geometry mappings across different environments. (2) Improved Training Stability: Unnormalized depth values can vary by orders of magnitude across scenes, leading to unstable gradients during training. Normalization mitigates this issue by providing a bounded input space.

# E    MORE DETAILS OF TRAINING AND INFERENCE

## E.1    GPU MEMORY REQUIREMENTS FOR MULTI-VIEW PROCESSING

We evaluate the GPU memory requirements on a single H20 GPU with input resolution of $518 \times 378$. The results are shown in Table 16.

Table 16: GPU memory consumption (GB) for different prediction tasks across varying numbers of input views. Measurements are performed on a single H20 GPU with resolution $518 \times 378$.

| N Views | Camera | Pointmap | Depth | Normal | 3DGS |
|---|---|---|---|---|---|
| 1 | 5.441 | 4.759 | 4.759 | 4.759 | 4.761 |
| 4 | 5.658 | 4.976 | 4.976 | 4.976 | 4.989 |
| 16 | 6.512 | 5.988 | 5.976 | 5.988 | 7.588 |
| 64 | 9.960 | 9.277 | 9.277 | 9.277 | 17.816 |
| 256 | 23.732 | 23.050 | 23.050 | 23.050 | 60.540 |
| 512 | 42.102 | 41.421 | 41.421 | 41.421 | OOM |

The memory consumption scales approximately linearly with the number of input views for most prediction tasks (camera, pointmap, depth, normal), growing from ∼5GB for single-view to ∼23GB for 256 views. However, the 3D Gaussian Splatting task exhibits significantly higher memory usage (60.5GB for 256 views), primarily due to additional convolutional layers in the GS head that decode high-dimensional Gaussian attributes (position, rotation, scaling, opacity, spherical harmonics) from dense feature maps. Further memory reduction could be achieved through optimized attention mechanisms and more compact 3D representations, following recent approaches like FastVGGT (Shen et al., 2025) and ReSplat (Xu et al., 2025a), which we leave for future work.

## E.2    MAXIMUM NUMBER OF INPUT VIEWS

We evaluate the maximum number of input views that our model can handle on a single H20 GPU with input resolution of $518 \times 378$. The results are shown in Table 17.

Table 17: Maximum number of input views supported by different prediction tasks on a single H20 GPU with resolution $518 \times 378$.

| Task | Camera | Pointmap | Depth | Normal | 3DGS |
|---|---|---|---|---|---|
| Max Views | 1024 | 1024 | 1024 | 1024 | 360 |

Most dense prediction tasks (camera, pointmap, depth, normal) can handle up to 1024 input views, while 3D Gaussian Splatting is limited to 360 views due to its higher memory footprint from decoding high-dimensional Gaussian attributes.

## E.3    TRAINING TIME (WALL-CLOCK)

We provide the wall-clock training time for full transparency and reproducibility. Our training was conducted on 32 NVIDIA H20 GPUs with gradient checkpointing for memory efficiency, Flash Attention v3 for accelerated attention computation, and mixed precision training (BF16).

The complete training process consists of two sequential stages. Stage 1 trains the dense prediction heads and requires approximately 42 hours of wall-clock time. Stage 2 trains the 3D Gaussian Splatting head and takes approximately 28 hours. The total end-to-end training time amounts to approximately 70 hours. Further acceleration could be achieved through advanced optimization techniques such as model parallelism (e.g., FSDP), FP8 precision training, or compiler-level optimizations (e.g., torch.compile).

# F    LIMITATIONS AND FUTURE WORKS

Despite the promising results achieved by our approach, several limitations remain. First, our method demonstrates suboptimal performance on dynamic scenes and autonomous driving environments, primarily due to the under-representation of such data in our training distribution. We plan

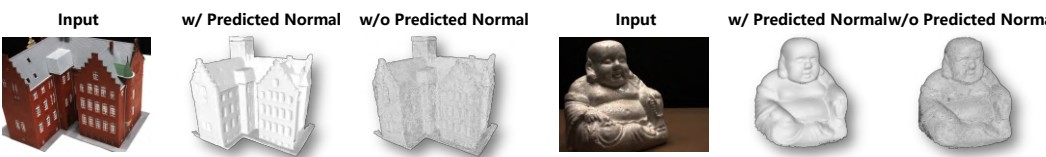

**Figure 7:** ***WorldMirror*** **Improves Surface Reconstruction with Predicted Normal Maps.**

to address this through strategic dataset expansion to enhance model generalization. Additionally, our current implementation supports input resolutions ranging from 300 to 700 pixels and cannot effectively handle scenarios where the number of input views reaches into the thousands. This constraint becomes particularly apparent when running on consumer-grade GPUs. Future work will explore computational optimizations to improve model efficiency and enable processing of longer visual sequences with reduced memory requirements.

## G  MORE VISUAL RESULTS

### G.1  NOVEL VIEW SYNTHESIS

In Fig. 8, we present additional results of feedforward Gaussians and their corresponding novel view renderings. Whether the input consists of AI-generated videos or real multi-view images, our method consistently infers 3D Gaussian splatting with plausible geometric structures and renders high-quality novel view images. This demonstrates that our model generalizes effectively across diverse input scenarios.

### G.2  POINT MAP RECONSTRUCTION

We provide additional visual comparisons of point map reconstruction in Fig. 9 and Fig. 10. Fig. 9 features selected scenes from 7-scenes, NRGBD, and DTU datasets, where comparisons with ground truth reveal that *WorldMirror* produces more consistent reconstructions, particularly when processing sparse viewpoints that require inference of spatial distributions. In Fig. 10, we evaluate model performance on in-the-wild images by processing both video generation model outputs and real-world multi-view captures. The results demonstrate that *WorldMirror* generates geometrically coherent and plausible reconstructions across these diverse inputs, highlighting its strong generalization capabilities.

## H  OTHER APPLICATIONS

### H.1  SURFACE RECONSTRUCTION.

*WorldMirror* supports high-quality 3D surface reconstruction with the predicted smooth normal maps. As shown in Fig. 7, by leveraging the predicted normals instead of traditional geometric normal estimation from point clouds, *WorldMirror* produces a cleaner surface with sharp details via Poisson surface reconstruction (Kazhdan et al., 2006).

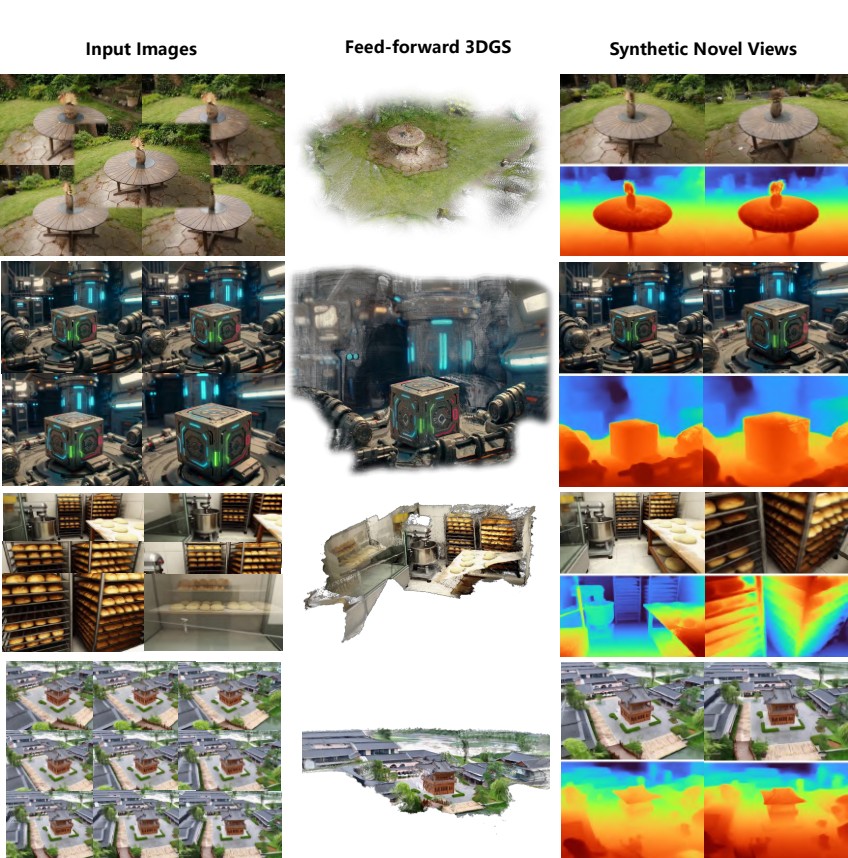

Figure 8: **Visual Results of Feed-Forward 3D Gaussians Generated by *WorldMirror*.**

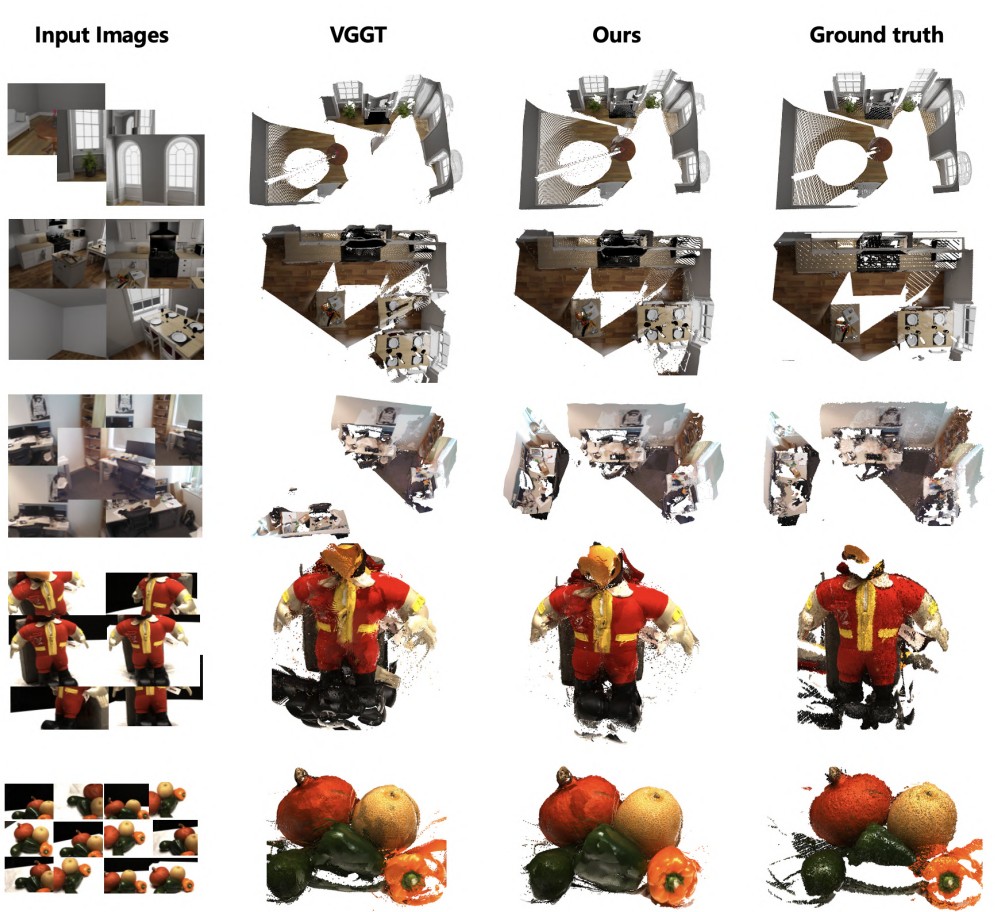

Figure 9: **Visual Comparisons on 7-Scenes, NRGBD, and DTU datasets.** *WorldMirror* delivers superior reconstruction fidelity compared to VGGT, effectively capturing spatial relationships within scenes while producing geometrically coherent structures.

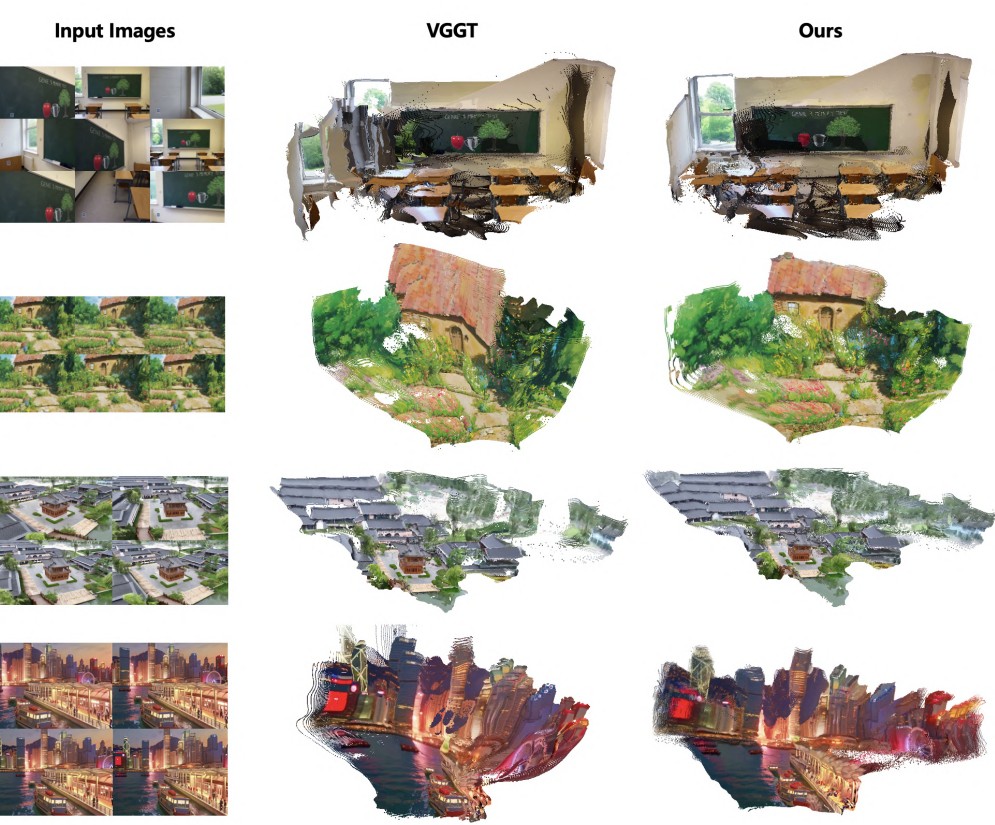

Figure 10: **Visual Comparisons of In-The-Wild Multi-View 3D Reconstruction.** *WorldMirror* delivers superior reconstruction fidelity with in-the-wild images as input, generating more plausible results in challenging scenarios compared to VGGT. Our approach effectively resolves complex spatial arrangements and maintains geometric consistency even when confronted with difficult viewing conditions, occlusions, or intricate environmental structures.

