# OpenReview forum: "WorldMirror: Universal  3D World Reconstruction with Any-Prior Prompting"
_ICLR.cc/2026/Conference — Submitted to ICLR 2026_

### Official Review · Reviewer_7G91 · 2025-10-30

**Soundness:** 3
**Presentation:** 3
**Contribution:** 3
**Rating:** 6
**Confidence:** 3

**Summary:**

This paper presents WorldMirror, a multi-view backbone capable of handling multiple input and output modalities. The potential inputs include images, camera intrinsics and extrinsics, and depth maps, while the outputs encompass pointmaps, surface normals, multi-view depths, camera parameters, and 3D Gaussians. The proposed model outperforms prior baselines that are trained for specific tasks, and experimental results demonstrate that incorporating additional input modalities consistently improves prediction accuracy.

**Strengths:**

1. The paper writing is clear, and the figures are visually appealing.
2. The proposed method supports unified inputs and outputs, which can be applied to a broader range of applications than previous methods taking only images as input.
3. The experiments across various domains comprehensively validate the effectiveness of the proposed method.

**Weaknesses:**

1. Using addition for image-like inputs (e.g., depth) might be suboptimal. A straightforward alternative is concatenation along the token dimension, though it may incur higher computational cost. Could the authors evaluate this trade-off?
2. The use of bolding and underlining in the tables is confusing. Conventionally, the best results are bolded and the second-best are underlined. However, in Table 1, WorldMirror (w/ intrinsics) with 0.042 is underlined for the mean accuracy on 7-Scenes, even though it is not the second-best result. Similarly, in Table 6, Ours with 20.29 is bolded for PSNR on RealEstate10K, despite not being the best value in that column.

**Questions:**

1. For depth map conditioning, the depth maps are normalized to the range [0, 1], which removes absolute scale information. Could this lead to issues or degrade performance?
2. In the appendix, the number of training epochs is listed. How long does the training take in terms of wall-clock time?

---

> ### Author Response · Authors · 2025-11-24
> **Response to Reviewer 7G91**
>
> Thank you for your valuable feedback to help us improve our paper. We have revised our paper based on your feedback. We detail our response below and please kindly let us know if our response addresses your concerns.
>
> ---
>
> > **Q1:** Using addition for image-like inputs (e.g., depth) might be suboptimal. A straightforward alternative is concatenation along the token dimension, though it may incur higher computational cost. Could the authors evaluate this trade-off?
>
> **A1:** Thank you for this insightful question. We compared concatenation (along token dimension) vs. addition (to image tokens) for depth prior injection:
>
> **Table R1:** Comparison of concatenation vs. addition for depth prior injection.
>
> | Depth Input       | 7S-Acc. ↓ | 7S-Comp. ↓ | DTU-Acc. ↓ | DTU-Comp. ↓ | Inference Time (s) ↓ | TFLOPs ↓|
> |:------------------|:----------|:-----------|:---------|:---------|:-----------|:------------|
> | Concat | 0.06 | 0.07 | 0.82 | 0.98 | 0.11 |  1.74 |
> | Addition (Ours)   | 0.06      | 0.06       | 0.79    | 0.97     |  0.09 |   1.14 |
>
> *Note: Metrics measured on a single 518×378 image on one H20 GPU.*
>
> Addition achieves better accuracy with 52.6% lower computational cost. This is because addition maintains pixel-wise correspondence between depth and image features while avoiding expensive attention operations. Details are added to Section D.1.
>
>
> > **Q2:** The use of bolding and underlining in the tables is confusing. Conventionally, the best results are bolded and the second-best are underlined. However, in Table 1, WorldMirror (w/ intrinsics) with 0.042 is underlined for the mean accuracy on 7-Scenes, even though it is not the second-best result. Similarly, in Table 6, Ours with 20.29 is bolded for PSNR on RealEstate10K, despite not being the best value in that column.
>
> **A2:** We apologize for the confusion. We have corrected the formatting: in Table 1 (7-Scenes), 0.042 and 0.014 should not be underlined; in Table 6 (RealEstate10K), our method (20.29) should be underlined (2nd best), not bolded. We have carefully reviewed and corrected all table formatting issues in the revised manuscript.
>
>
> > **Q3:** For depth map conditioning, the depth maps are normalized to the range [0, 1], which removes absolute scale information. Could this lead to issues or degrade performance?
>
> **A3:** Thank you for this question. Yes, depth normalization to [0,1] removes absolute scale, but this is by design. WorldMirror predicts geometry in relative scale (like COLMAP, VGGT), not absolute metric scale. Our ablation study on 7-Scenes and DTU shows normalized depth actually improves performance:
>
> **Table R2:** Comparison of normalized vs. unnormalized depth input on 7-Scenes and DTU datasets.
>
> | Depth Input       | 7S-Acc. ↓ | 7S-Comp. ↓ | 7S-NC1 ↑ | 7S-NC2 ↑ | DTU-Acc. ↓ | DTU-Comp. ↓ | DTU-NC1 ↑ | DTU-NC2 ↑ |
> |:------------------|:----------|:-----------|:---------|:---------|:-----------|:------------|:----------|:----------|
> | Unnormalized | 0.11      | 0.12       | 0.66     | 0.67     | 2.60       | 5.96        | 0.52      | 0.51      |
> | Normalized (Ours)   | 0.06      | 0.06       | 0.77     | 0.78     | 0.79       | 0.97        | 0.70      | 0.71      |
>
> The significant improvement demonstrates normalized depth provides more stable and consistent features for relative-scale reconstruction. Details are added to Section D.2.
>
> > **Q4:** In the appendix, the number of training epochs is listed. How long does the training take in terms of wall-clock time?
>
> **A4:** Thank you for this question. Our training uses 32 NVIDIA H20 GPUs with gradient checkpointing, Flash Attention, and mixed precision (BF16). Stage 1 (dense prediction heads) takes ~42 hours, Stage 2 (3D Gaussian Splatting head) takes ~28 hours, for a **total of ~70 hours**. Further acceleration is possible via model parallelism (FSDP), FP8 precision, or compiler optimizations (torch.compile). Details are added to Section E.3.

---

### Official Review · Reviewer_q977 · 2025-10-31

**Soundness:** 3
**Presentation:** 3
**Contribution:** 3
**Rating:** 4
**Confidence:** 4

**Summary:**

This paper presents WorldMirror, a unified feed-forward model for 3D reconstruction that addresses two key limitations of existing methods. First, it introduces a multi-modal prior prompting mechanism that flexibly incorporates camera poses, intrinsics, and depth maps alongside images. Second, it unifies multiple tasks (point cloud reconstruction, camera estimation, depth prediction, normal estimation, and novel view synthesis) within a single architecture. The authors employ curriculum learning across task sequencing, data scheduling, and resolution to train this multi-task model effectively. Experiments demonstrate state-of-the-art performance across diverse benchmarks, with significant improvements when priors are available.

**Strengths:**

- The paper is clearly written and easy to follow.

- The multi-modal prior prompting approach is well-designed, with specialized encoding strategies for different modalities (single tokens for compact representations vs. dense tokens for spatial information).

- The ability to leverage any available priors while maintaining feed-forward efficiency addresses real-world scenarios where auxiliary information is often accessible, with demonstrated improvements.

**Weaknesses:**

- The training procedure is complex (15 datasets, multi-stage curriculum learning), which raises reproducibility concerns.

- The paper lacks a comparison with other methods that can accept auxiliary 3D information (e.g., Pow3R [1] and MapAnything [2]).

[1] Pow3r: Empowering unconstrained 3d reconstruction with camera and scene priors

[2] MapAnything: Universal Feed-Forward Metric 3D Reconstruction

**Questions:**

While the method accepts various priors, the paper assumes they are accurate and provides insufficient analysis of robustness to noisy or low-quality priors. In real-world scenarios, obtaining high-quality camera poses or depth from LiDAR/RGB-D sensors is often challenging, yet the paper doesn't evaluate degradation under realistic noise conditions.

---

> ### Author Response · Authors · 2025-11-24
> **Response to Reviewer q977 (1/2)**
>
> Thank you for your valuable feedback to help us improve our paper. We have revised our paper based on your feedback. We detail our response below and please kindly let us know if our response addresses your concerns.
>
> ---
>
> > **Q1:** The training procedure is complex (15 datasets, multi-stage curriculum learning), which raises reproducibility concerns.
>
> **A1:** Thank you for raising this concern. While we use 15 datasets, the training is straightforward with two stages: Stage 1 jointly trains all dense prediction heads (camera, pointmap, depth, normal) for robust multi-task features; Stage 2 freezes these heads and trains only the 3D Gaussian Splatting head for photo-realistic rendering.
>
> To ensure reproducibility, we will publicly release: (1) complete training configurations (hyperparameters, learning rates, batch sizes, optimization settings); (2) data preprocessing scripts and dataset mixing strategies; (3) curriculum learning schedules with epoch ranges and task weights; (4) pre-trained checkpoints for both stages. All details are in Appendix A.2, and code/configs/checkpoints will be made publicly available.
>
> > **Q2:** The paper lacks a comparison with other methods that can accept auxiliary 3D information (e.g., Pow3R [1] and MapAnything [2]).
>
> **A2:** Thank you for this suggestion. We compare with Pow3R [1] and MapAnything [2] under different prior conditions. Since Pow3R is designed for 2-view reconstruction, we extend it to multi-view using Procrustes alignment for camera pose estimation:
>
> **Table R1:** Comparison with Pow3R and MapAnything under different prior conditions on 7-Scenes, NRGBD, and DTU datasets.
>
> | Method | Prior Condition | 7-Scenes Acc ↓ | 7-Scenes Comp ↓ | NRGBD Acc ↓ | NRGBD Comp ↓ | DTU Acc ↓ | DTU Comp ↓ |
> |:-------|:----------------|:---------------|:----------------|:------------|:-------------|:----------|:-----------|
> | Pow3R(pro)  | None            | 0.103          | 0.174           | 0.121       | 0.099        | 5.104     | 2.863      |
> | MapAnything | None       | 0.075          | 0.093           | 0.088       | 0.100        | 1.997     | 2.068      |
> | WorldMirror | None | **0.044**  | **0.050**       | **0.038**   | **0.041**    | **0.982** | **1.486**  |
> |        |                 |                |                 |             |              |           |            |
> | Pow3R(pro)  | Intrinsics      | 0.104          | 0.175           | 0.120       | 0.101        | 4.336     | 2.711      |
> | MapAnything | Intrinsics | 0.073          | 0.090           | 0.086       | 0.100        | 2.309     | 2.011      |
> | WorldMirror | Intrinsics | **0.048** | **0.051**    | **0.038**   | **0.042**    | **0.948** | **1.579**  |
> |        |                 |                |                 |             |              |           |            |
> | Pow3R(pro)  | Depth           | 0.103          | 0.174           | 0.121       | 0.099        | 5.104     | 2.863      |
> | MapAnything | Depth      | 0.067          | 0.080           | 0.065       | 0.073        | 3.879     | 2.403      |
> | WorldMirror | Depth | **0.058**  | **0.060**       | **0.028**   | **0.027**    | **0.790** | **0.977**  |
> |        |                 |                |                 |             |              |           |            |
> | Pow3R(pro)  | Camera Pose     | 0.049          | 0.049           | 0.074       | 0.062        | 4.342     | 2.465      |
> | MapAnything | Camera Pose | 0.029         | **0.032**           | 0.047       | 0.045        | 2.394     | 2.073      |
> | WorldMirror | Camera Pose | **0.023** | 0.035   | **0.023**   | **0.026**    | **0.967** | **1.502**  |
> |        |                 |                |                 |             |              |           |            |
> | Pow3R(pro)  | All Priors      | 0.049          | 0.046           | 0.072       | 0.060        | 3.526     | 2.309      |
> | MapAnything | All Priors | **0.012**          | **0.013**           | 0.016       | 0.013        | 1.707     | **0.989**      |
> | WorldMirror | All Priors | 0.018 | 0.024   | **0.013**   | **0.011**    | **0.717** | 0.947  |
>
>   *Note: Pow3R(pro) refers to Pow3R with procrustes alignment.*
>
>
> WorldMirror consistently outperforms both methods across all prior conditions, due to cleaner multi-view-friendly embedding strategies for intrinsics and poses, and better geometric consistency across views. These comparisons are added as Table 5 in the revised manuscript.
>
> [1] Pow3r: Empowering unconstrained 3d reconstruction with camera and scene priors
> [2] MapAnything: Universal Feed-Forward Metric 3D Reconstruction

---

> ### Author Response · Authors · 2025-11-24
> **Response to Reviewer q977 (2/2)**
>
> > **Q3:** While the method accepts various priors, the paper assumes they are accurate and provides insufficient analysis of robustness to noisy or low-quality priors. In real-world scenarios, obtaining high-quality camera poses or depth from LiDAR/RGB-D sensors is often challenging, yet the paper doesn't evaluate degradation under realistic noise conditions.
>
> **A3:** Thank you for this important question. We evaluate robustness to noisy priors by injecting controlled noise following Pow3R [1] to simulate real-world sensor errors:
>
> **Noise settings:** Camera pose with rotation noise (0°/clean, 20°, 40°) and translation noise (Gaussian σ=0.0/clean, 0.05, 0.1); intrinsics scaled by 1.0×/clean, 0.5×, 0.1×; depth with multiplicative noise ~ N(1.0, σ) where σ=0.0/clean, 0.05, 0.1.
>
> Results on 7-Scenes and DTU (compared to no-prior baseline):
>
> **Table R2:** Robustness evaluation with noisy priors on 7-Scenes and DTU datasets.
>
> | Prior Type | Noise Level | 7-Scenes Acc ↓ | 7-Scenes Comp ↓ | DTU Acc ↓ | DTU Comp ↓ |
> |:-----------|:------------|:---------------|:----------------|:----------|:-----------|
> | None (baseline) | - | 0.044 | 0.050 | 0.982 | 1.486 |
> |  |  |  |  |  |  |
> |  Pose (Rotation) | 0° (clean) | **0.022** | **0.035** | **0.966** | 1.502 |
> |  | 20° | 0.023 | **0.035** | 0.976 | 1.483 |
> |  | 40° | 0.025 | 0.036 | 1.017 | **1.466** |
> |  |  |  |  |  |  |
> |  Pose (Translation) | σ=0.0 (clean) | **0.022** | **0.035** | **0.966** | 1.502 |
> |  | σ=0.05 | **0.022** | **0.035** | 0.967 | 1.503 |
> |  | σ=0.1 | 0.024 | 0.036 | 0.967 | 1.503 |
> |  |  |  |  |  |  |
> | Intrinsics | 1.0× (clean) | 0.047 | 0.051 | **0.948** | **1.579** |
> |  | 0.8× | **0.044** | **0.049** | 1.047 | 1.740 |
> |  | 0.6× | 0.051 | 0.054 | 2.691 | 1.830 |
> | Depth | σ=0.0 (clean) | **0.058** | **0.060** | **0.790** | **0.977** |
> |  | σ=0.05 | **0.058** | 0.061 | 0.795 | 0.980 |
> |  | σ=0.1 | 0.064 | 0.071 | 1.387 | 1.820 |
>
>
>
> **Key findings:** (1) **Graceful degradation**: Performance degrades smoothly as noise increases, indicating robust feature learning. (2) **Benefits persist**: Even with moderate noise (20° rotation, 0.5× intrinsics, σ=0.05 depth), noisy priors still outperform the no-prior baseline. (3) **Pose robustness**: Camera pose priors remain robust up to 20° rotation error. (4) **Depth sensitivity**: Depth priors are more sensitive to noise but maintain reasonable performance with σ=0.05.
>
> These results demonstrate WorldMirror effectively leverages imperfect priors with robustness to realistic noise levels. Details are added to Appendix C.

---

### Official Review · Reviewer_9nA6 · 2025-11-01

**Soundness:** 3
**Presentation:** 3
**Contribution:** 2
**Rating:** 6
**Confidence:** 3

**Summary:**

This paper introduces a unified neural network architecture designed for a variety of 3D reconstruction tasks. The inputs can take multiple optional forms, such as RGB images, depth maps, and camera intrinsic or extrinsic parameters. The outputs include point maps, depth maps, surface normals, camera parameters, and 3D Gaussians. Overall, it presents a versatile and general framework. The reported experiments demonstrate impressive quantitative and qualitative results across multiple tasks. This appears to be a large-scale and ambitious project, and I appreciate the significant engineering effort invested by the authors.

In summary, however, the main idea somewhat reiterates a well-known observation. That jointly training on multiple related tasks can improve performance across them. This insight, while valid, is not particularly novel as it has been established in many prior works. See details below.

Nonetheless, the experimental results are strong, and the implementation and training complexity reflect a commendable level of technical contribution. Overall, I feel this paper sits at the borderline of acceptance.

**Strengths:**

- This work requires a substantial amount of experimentation and engineering effort, which I truly appreciate.

- The experimental results are strong, demonstrating state-of-the-art performance across several tasks such as depth estimation, normal estimation, and novel view synthesis.

- The visualization results are also clear and compelling.

**Weaknesses:**

The main idea of unifying multiple related tasks into a single framework to achieve mutual performance improvement is not novel — it has been explored and confirmed in numerous prior studies. For example:

- [CVPR 2018] “Multi-Task Learning Using Uncertainty to Weigh Losses for Scene Geometry and Semantics” jointly learns depth, semantic segmentation, and instance segmentation, achieving better performance than single-task models.

- [CVPR 2018] “Taskonomy: Disentangling task transfer learning” provides a principled framework and empirical evidence showing that related tasks can effectively transfer and boost performance for each other.

- [CVPR 2020]: “Pattern-structure diffusion for multi-task learning” jointly trains depth, segmentation, and surface normals on NYUD-v2 and SUN RGB-D, showing consistent performance gains through multi-task learning.

Notably, the above papers are not cited in this work. There are likely additional relevant studies as well.

I suggest that the authors expand the Related Work section to better discuss how this paper differs from these prior efforts, and to clarify whether it brings any new conclusions or insights that might be of particular interest to the community.

**Questions:**

What is the GPU memory requirement for processing N images in each prediction task?

---

> ### Author Response · Authors · 2025-11-24
> **Response to Reviewer 9nA6**
>
> Thank you for your valuable feedback to help us improve our paper. We have revised our paper based on your feedback. We detail our response below and please kindly let us know if our response addresses your concerns.
>
> ---
>
> > **Q1:** The main idea of unifying multiple related tasks into a single framework to achieve mutual performance improvement is not novel — it has been explored and confirmed in numerous prior studies. Notably, the above papers are not cited in this work. There are likely additional relevant studies as well.
>
> **A1:** We sincerely thank the reviewer for the valuable feedback. We have added the suggested works (Uncertainty weighting, Taskonomy, Pattern-structure diffusion) and other related works in Section 2. While multi-task learning is well-established, our contribution lies in **two novel aspects** beyond general MTL:
>
> **Difference #1: Task-specific innovations.** We introduce novel designs for individual tasks (Section 3.2): hybrid supervision for normal estimation, and personalized depth architecture with novel view supervision for 3DGS. These are independent technical contributions beyond applying existing MTL methods.
>
> **Difference #2: Unique 3D multi-task training dynamics.** Direct joint-training of 3DGS heads with dense prediction heads causes conflicts—GS optimizes for photo-realistic rendering while others optimize for precise geometry. This conflict is not observed in 2D MTL works. We address this via curriculum learning (Section A.2): first training all tasks except GS for robust features, then freezing dense heads and training only the GS head.
>
> > **Q2:** What is the GPU memory requirement for processing N images in each prediction task?
>
> **A2:** Thank you for this question. We evaluate GPU memory on a single H20 GPU with 518×378 resolution:
>
> **Table R1:** GPU memory consumption for different tasks with varying number of input views.
>
> | N views | Camera   | Pointmap | Depth    | Normal   | 3DGS     |
> |:--------|:---------|:---------|:---------|:---------|:---------|
> | 1       | 5.441GB  | 4.759GB  | 4.759GB  | 4.759GB  | 4.761GB  |
> | 4       | 5.658GB  | 4.976GB  | 4.976GB  | 4.976GB  | 4.989GB  |
> | 16      | 6.512GB  | 5.988GB  | 5.976GB  | 5.988GB  | 7.588GB  |
> | 64      | 9.960GB  | 9.277GB  | 9.277GB  | 9.277GB  | 17.816GB |
> | 256     | 23.732GB | 23.050GB | 23.050GB | 23.050GB | 60.540GB |
> | 512     | 42.102GB | 41.421GB | 41.421GB | 41.421GB | OOM |
>
> Memory scales approximately linearly for most tasks (camera, pointmap, depth, normal), growing from ~5GB for single-view to ~23GB for 256 views. However, 3DGS requires significantly more memory (60.5GB for 256 views) due to convolutional layers that decode high-dimensional Gaussian attributes (position, rotation, scaling, opacity, spherical harmonics) from dense features. Further reduction is possible via optimized attention and compact representations [1,2], which we leave for future work. Details are in Section E.1.
>
> [1] FastVGGT: Training-Free Acceleration of Visual Geometry Transformer
> [2] ReSplat: Learning Recurrent Gaussian Splats

---

### Official Review · Reviewer_9c7Q · 2025-11-01

**Soundness:** 3
**Presentation:** 3
**Contribution:** 2
**Rating:** 4
**Confidence:** 5

**Summary:**

This paper presents WorldMirror, a feedforward geometric model that supports various priors (intrinsic, pose, and depth) and multiple tasks, including camera pose estimation, normal estimation, point cloud reconstruction, and novel view synthesis. The model is built upon VGGT, extending it with additional task-specific heads and priors to support a broader range of inputs and objectives. Experimental results demonstrate that the proposed model achieves state-of-the-art performance across multiple tasks.

**Strengths:**

- The proposed model achieves SOTA performance on several tasks, including pose estimation, point prediction, normal prediction, and novel view synthesis.

- The model supports multiple inputs and tasks within a single unified framework.

**Weaknesses:**

- Some parts of the paper closely resemble previous works but lack proper attribution. For example, the proposed Multi-Modal Prior Prompting is similar to Pow3r, except adapted to a multi-view setting. In Line 75, the paper states “we propose”, but the same training strategy has already been explored in Pow3r. Overall, the proposed model appears to be a combination of VGGT, Pow3r, and AnySplat.

- For the novel view synthesis task, the experiments are restricted to 64 input views, whereas AnySplat reports results on over 100 views and compares performance with optimization-based methods (e.g., original 3DGS).

- The paper lacks performance comparisons with recent SOTA novel view synthesis models such as DepthSplat.

**Questions:**

- What is the input resolution used during evaluation, and how does it compare with previous methods across different tasks, given that different models may use different input resolutions?

- For the two-view novel view synthesis setting, how does the proposed method compare with NoPoSplat?

- What is the maximum number of input views that the model can handle?

---

> ### Author Response · Authors · 2025-11-24
> **Response to Reviewer 9c7Q (1/3)**
>
> Thank you for your valuable feedback to help us improve our paper. We have revised our paper based on your feedback. We detail our response below and please kindly let us know if our response addresses your concerns.
>
> ---
>
> > **Q1:** Some parts of the paper closely resemble previous works but lack proper attribution. For example, the proposed Multi-Modal Prior Prompting is similar to Pow3r, except adapted to a multi-view setting. In Line 75, the paper states "we propose", but the same training strategy has already been explored in Pow3r. Overall, the proposed model appears to be a combination of VGGT, Pow3r, and AnySplat.
>
> **A1:** Thank you for this constructive feedback. We have revised the manuscript to acknowledge related works and highlight our novel contributions.
>
> **Our Unified Framework and Key Contributions:** WorldMirror is the first framework unifying both inputs (multiple geometric priors: intrinsics, pose, depth) and outputs (five tasks: camera, pointmap, depth, normal, NVS). To address conflicting optimization objectives, we propose a two-stage curriculum learning: first training dense prediction heads for robust geometric features, then training the 3DGS head for photo-realistic rendering. Other specific technical innovations includes:
>
> **Multi-modal Prior Embedding vs. Pow3R [1]:** We acknowledge that Pow3R pioneered incorporating geometric priors for two-view reconstruction. However, our Multi-Modal Prior Prompting differs:
> - **Better prior embedding:** Unlike Pow3R's two-view-specific injection (dense ray map embeddings for intrinsics in encoder, poses in decoder), we use a simpler unified approach: directly concatenating intrinsics and pose priors as tokens with image tokens at input. Table 7 shows our token-based strategy achieves 3.57% better performance with 75.2% fewer parameters.
> - **Multi-view extension with cross-view consistency:** Pow3R processes two views independently, while our approach handles multi-view scenarios with explicit cross-view geometric consistency through attention mechanisms.
>
> **Universal Geometric Prediction vs. AnySplat [2]:** While we build upon VGGT's [3] architecture and extend it to support novel view synthesis similar to AnySplat, our key contributions differ:
> - **Unified multi-task framework:** Unlike VGGT (camera + geometry) and AnySplat (NVS only), we unify five tasks in a single model with joint geometric reasoning.
> - **Novel NVS architecture design:** We introduce personalized geometry architecture and novel view supervision for feed-forward 3DGS, which are independent contributions beyond AnySplat. Table 8 shows our strategies improve NVS: dedicated GS-DPT head (+1.49%) and novel-view supervision (+6.28%).
> - **Prior-conditioned inference:** We enable flexible conditioning on various geometric priors during inference, which neither VGGT nor AnySplat support.
>
>
>
> [1] Pow3r: Empowering unconstrained 3d reconstruction with camera and scene priors
> [2] AnySplat: Large-Scale Generalizable 3D Gaussian Splatting
> [3] VGGT: Visual Geometry Grounded Transformer
>
> > **Q2:** For the novel view synthesis task, the experiments are restricted to 64 input views, whereas AnySplat reports results on over 100 views and compares performance with optimization-based methods (e.g., original 3DGS).
>
> **A2:** Thank you for highlighting this point. We have added experiments with more than 100 input views and included comparisons with optimization-based methods.
> We evaluate novel view synthesis with 100, 150, and 200 input views on the MatrixCity dataset [1]. The results are shown in the table below:
>
> **Table R1:** Novel view synthesis performance with 100, 150, and 200 input views on MatrixCity dataset.
>
> | Method | PSNR | SSIM | LPIPS | PSNR | SSIM | LPIPS | PSNR | SSIM | LPIPS |
> |--------|------|------|-------|------|------|-------|------|------|-------|
> | | **100 views** | | | **150 views** | | | **200 views** | | |
> | 3D-GS | 18.21 | 0.568 | 0.445 | 18.86 | 0.593 | 0.412 | 19.54 | 0.612 | 0.388 |
> | Mip-Splatting | 17.97 | 0.536 | 0.450 | 18.24 | 0.579 | 0.438 | 18.63 | 0.588 | 0.414 |
> | AnySplat | 20.51 | 0.620 | **0.347** | 19.24 | 0.601 | 0.399 | 19.18 | 0.605 | 0.397 |
> | WorldMirror (Ours) | **20.88** | **0.640** | 0.360 | **20.62** | **0.626** | **0.370** | **20.36** | **0.630** | **0.375** |
>
> These experiments demonstrate that our method generalizes far beyond the maximum of 24 input views used during training, and achieves superior performance to both feed-forward and optimization-based approaches, without any post-processing or additional refinement. Details are added to Section B.3.
>
> [1] MatrixCity: A Large-scale City Dataset for City-scale Neural Rendering and Beyond

---

> ### Author Response · Authors · 2025-11-24
> **Response to Reviewer 9c7Q (2/3)**
>
> > **Q3:** The paper lacks performance comparisons with recent SOTA novel view synthesis models such as DepthSplat.
>
> **A3:** Thank you for pointing out the absence of comparisons with previous NVS methods. We have added a performance comparison between our approach and DepthSplat [1] on DL3DV. The results are shown in the table below:
>
> **Table R2:** Comparison with DepthSplat and other methods on DL3DV with different prior settings.
>
> | Method | Pose | Intrinsic | Resolution | PSNR | SSIM | LPIPS | PSNR | SSIM | LPIPS | PSNR | SSIM | LPIPS |
> |--------|------|-----------|------------|------|------|-------|------|------|-------|------|------|-------|
> | | | | | **8 views** | | | **24 views** | | | **64 views** | | |
> | AnySplat | | | 448x252 | 15.62 | 0.453 | 0.305 | 18.68 | 0.571 | **0.221** | 19.50 | 0.610 | **0.208** |
> | WorldMirror (Ours) | | | 448x252 | **17.50** | **0.518** | **0.303** | **19.15** | **0.602** | 0.241 | **19.51** | **0.626** | 0.239 |
> | | | | | | | | | | | | | |
> | FLARE | | ✓ | 256x256 | 14.77 | 0.412 | 0.647 | 14.11 | 0.398 | 0.761 | - | - | - |
> | WorldMirror (Ours) | | ✓ | 252x252 | **16.83** | **0.480** | **0.320** | **18.76** | **0.574** | **0.244** | **19.10** | **0.593** | **0.240** |
> | | | | | | | | | | | | | |
> | DepthSplat | ✓ | ✓ | 448x256 | 18.79 | 0.619 | 0.316 | 18.71 | 0.643 | 0.313 | 16.80 | 0.551 | 0.416 |
> | WorldMirror (Ours) | ✓ | ✓ | 448x252 | **19.08** | **0.624** | **0.261** | **20.24** | **0.675** | **0.221** | **20.30** | **0.680** | **0.226** |
>
> As shown in the table, our method demonstrates significant advantages over DepthSplat:
>
> - **Consistent performance improvements across all settings**, with particularly notable gains in 24-view (+1.53 PSNR) and 64-view (+3.50 PSNR) scenarios.
>
> - **Superior scalability**: Unlike DepthSplat which suffers from performance degradation as input views increase, our method consistently improves with more views, demonstrating better capability for handling larger numbers of input images.
>
> - **Greater flexibility in prior configurations**: Our framework maintains strong performance even without pose or intrinsic information, while DepthSplat strictly requires both geometric priors.
>
> Details are added to Section 4.1 in L412.
>
> [1] DepthSplat: Connecting Gaussian Splatting and Depth
>
> > **Q4:** What is the input resolution used during evaluation, and how does it compare with previous methods across different tasks, given that different models may use different input resolutions?
>
> **A4:** Thank you for this question. We configure resolution settings to ensure fair comparisons while respecting each method's design.
>
> **Dense prediction tasks (camera, pointmap, depth, normal):** We follow established protocols [1,2]. Methods with patch size 16 (Fast3R, CUT3R, FLARE) use 512px long edge; methods with patch size 14 (VGGT, Pi3, WorldMirror) use 518px. Surface normal uses unified 640×480.
>
> **Novel view synthesis:** Main paper uses unified 518×376 for consistency. To ensure fairness, we additionally evaluate each baseline at its optimal resolution in A3. We have added these multi-resolution experiments to the Table 5. Our model is trained with **dynamic resolutions (100K-250K pixels)**, enabling robust generalization across various input resolutions while consistently outperforming baselines.
>
> [1] Pi3: Permutation-Equivariant Visual Geometry Learning
> [2] Rethinking Inductive Biases for Surface Normal Estimation

---

> ### Author Response · Authors · 2025-11-24
> **Response to Reviewer 9c7Q (3/3)**
>
> > **Q5:** For the two-view novel view synthesis setting, how does the proposed method compare with NoPoSplat?
>
> **A5:** Thank you for this suggestion. We conduct two-view NVS experiments on both RealEstate10K and DL3DV compared with NoPoSplat. The results are shown in the table below:
>
> **Table R3:** Two-view novel view synthesis comparison with NoPoSplat on RealEstate10K and DL3DV datasets.
>
> | Method | Intrinsic | PSNR | SSIM | LPIPS | PSNR | SSIM | LPIPS |
> |--------|-----------|------|------|-------|------|------|-------|
> | |  | **Re10K** | | | **DL3DV** | | |
> | NoPoSplat  | ✓ | **25.06** | **0.836** | 0.164 | 19.00 | 0.575 | 0.350 |
> | AnySplat |  | 18.01 | 0.602 | 0.207 | 13.56 | 0.368 | 0.338 |
> | WorldMirror (Ours)  | | 23.48 | 0.805 | 0.124 | 18.41 | 0.582 | 0.270 |
> | WorldMirror (Ours)  | ✓ | 23.89 | 0.826 | **0.113** | **19.08** | **0.636** | **0.250** |
>
> Notably, our model is **not** trained specifically on RealEstate10K for the two-view setting, yet it achieves performance **comparable to NoPoSplat** across most metrics. Moreover, when compared to AnySplat, which more closely matches our training configuration, our method **substantially outperforms** it. Details are added to Section B.4.
>
> > **Q6:** What is the maximum number of input views that the model can handle?
>
> **A6:** Thank you for this question. We evaluate the GPU memory requirements on a single H20 GPU with input resolution of 518×378. The results are shown in the table below:
>
> **Table R4:** Maximum number of input views that can be handled for different prediction tasks.
>
> |  | Camera   | Pointmap | Depth    | Normal   | 3DGS     |
> |:--------|:---------|:---------|:---------|:---------|:---------|
> | **N_views**       | 1024  | 1024 | 1024 | 1024  | 360  |
>
>
>  Further memory reduction could be achieved through optimized attention mechanisms and more compact 3D representations, following recent approaches like FastVGGT [1] and ReSplat [2], which we leave for future work. We have added these results in the revised manuscript (Section E.2).
>
> [1] FastVGGT: Training-Free Acceleration of Visual Geometry Transformer
> [2] ReSplat: Learning Recurrent Gaussian Splats

---

### Author Response · Authors · 2025-11-27

Dear reviewers,

We hope this message finds you well. As the discussion phase nears its end, we would greatly appreciate it if you could briefly review our response and revised paper at your convenience. We are happy to clarify any questions.

Thank you for your time and valuable feedback.

Best regards, Authors of Paper 4002

---

### Meta-Review · Area_Chair_Cceg · 2026-01-09

**Summary:**

This paper presents a unified feed-forward framework for multi-task 3D perception and reconstruction that incorporates various auxiliary priors and demonstrates strong empirical performance across multiple tasks, including pose estimation, depth and normal prediction, and novel view synthesis. The work clearly involves substantial engineering effort, large-scale experimentation across many datasets, and careful system design. The paper is also generally well written, and the qualitative and quantitative results are visually compelling.

However, despite these strengths, as raised by reiveiwers, AC do not find the core technical contributions sufficiently novel or clearly distinguished from prior work to warrant acceptance at this time.

First, the central idea of jointly learning multiple related geometric tasks within a single framework to achieve mutual performance gains is well established in the literature, with extensive prior work dating back several years (e.g., Taskonomy, uncertainty-weighted multi-task learning, pattern-structure diffusion). These foundational works are not cited, nor is it clearly articulated how this paper provides new conceptual insights beyond confirming known benefits of multi-task learning at a larger scale. After rebuttal, the authors added the related works, but the comparisons were not done properly.

Second, the proposed Multi-Modal Prior Prompting mechanism appears closely related to existing prior-driven reconstruction approaches, particularly Pow3r, with similar training strategies and design philosophies adapted to a multi-view setting. In multiple places, the paper claims novelty (“we propose”) without adequately acknowledging or differentiating from these prior methods. More broadly, the overall framework resembles a combination of existing components (e.g., VGGT-style feature extraction, Pow3r-like prior usage, AnySplat-style Gaussian-based rendering), but the paper does not convincingly articulate what fundamentally new principle or capability emerges from this combination.

Finally, the complexity of the training pipeline—involving many datasets and multi-stage curriculum learning—raises concerns about reproducibility and obscures which components are truly responsible for the reported gains. Some design choices (e.g., additive fusion of image-like priors) are not sufficiently justified or compared against reasonable alternatives. There are also minor but noticeable issues in result presentation (e.g., inconsistent bold/underline usage in tables), which further detract from clarity.

In summary, while the paper demonstrates strong empirical results and significant engineering effort, the lack of clear novelty, insufficient engagement with relevant prior work, and incomplete experimental comparisons substantially weaken its contribution. AC encourage the authors to more clearly position their work with respect to existing literature, strengthen comparative evaluations, and clarify the unique insights offered by the proposed framework in a future revision.

**Reviewer Concerns:**

For novel view synthesis, experiments are initially limited to 64 input views, while closely related methods such as AnySplat evaluate on significantly larger numbers of views and include comparisons with optimization-based 3D Gaussian Splatting. Additionally, comparisons with recent state-of-the-art methods such as DepthSplat were missing, and there was no direct comparison with other feed-forward methods that also accept auxiliary 3D priors (e.g., Pow3r, MapAnything). However, after revisions, these experiments were fully added.

**Reviewer Scores:**

They might be unchanged.

---

### Decision · Program_Chairs · 2026-01-26

Reject